# TWO INSTANCES OF INTERPRETABLE NEURAL NETWORK FOR UNIVERSAL APPROXIMATIONS

## ABSTRACT

This paper proposes two bottom-up interpretable neural network (NN) constructions for universal approximation, namely Triangularly-constructed NN (TNN) and Semi-Quantized Activation NN (SQANN). The notable properties are (1) resistance to catastrophic forgetting (2) existence of proof for arbitrarily high accuracies on training dataset (3) for an input $x$, users can identify specific samples of training data whose activation "fingerprints" are similar to that of $x$'s activations. Users can also identify samples that are out of distribution.

## 1 INTRODUCTION

Artificial neural networks (NN) have recently seen successful applications in many fields. Modern deep neural network (DNN) architecture, usually trained through the backpropagation mechanism, has been called a black-box because of its lack of interpretability. To tackle this issue, various studies have been performed to understand how a NN works; see the following surveys Arrieta et al. (2020); Gilpin et al. (2018); Tjoa & Guan (2020); Wiegreffe & Marasović (2021). This paper primarily proposes two interpretable models, namely triangularly-constructed NN (TNN) and Semi-Quantized Activation NN (SQANN). Both possess the following three notable properties: (1) Resistance to *catastrophic forgetting*. (2) Mathematical proofs for arbitrarily high accuracy on training datasets; experimentally demonstrable with python code and simple common datasets (see supp. materials). (3) Detection of out-of-distribution samples through weak activations.

**Concept disambiguation**. Several concepts have multiple possible definitions. We clarify the definitions used in this paper.

1. Interpretability. We consider only fine-grained interpretation, i.e. we look at the meaning of each single neuron or its activation in our models.

2. Universal approximation. Readers might be familiar with universal approximation of functions on certain conditions, e.g. compact sets. Our models can be more general, e.g. user can freely choose the interpolation function between 2 activations based on knowledge of the local manifold. The function can even be pathological. See appendix A.2.1.

3. Catastrophic forgetting: the tendency for knowledge of previously learned dataset to be abruptly lost as information relevant to a new dataset is incorporated. This definition is a slightly nuanced version of Kirkpatrick et al. (2017). Hence, our models' resistance to catastrophic forgetting is the following. Given a training dataset $D$ accurately modeled by an architecture $M$, a new dataset $D'$ (especially new, out of distribution dataset) can be incorporated into $M$ without losing accuracy on $D$. See appendix A.2.2 (analogy to biological system included).

**Related works and interpretability issues**. Recent remarkable studies on universal approximators include the Deep Narrow Network by Kidger & Lyons (2020), DeepONet universal approximation for operators by Lu et al. (2021) and the Broad Learning System by Chen et al. (2019); Hanin (2019); Park et al. (2021); Johnson (2019). While insightful, they do not directly address the eXplainable Artificial Intelligence (XAI) issue, especially the blackbox property of the DNN. Similarly, a number of classical papers provide theoretical insights for NN as universal approximators, but interpretability, transparency and fairness issues are not their main focus. The universal approximation theorem by Cybenko (1989) asserts that a NN with a single hidden layer can approximate any function to arbitrarily small error under common conditions, proven by asserting the density of that set of NN

in the function space using classic mathematical theorems. In particular, its theorem 1 uses an abstract proof by contradiction. From the proof, it is not easy to observe the internal mechanism of a NN in a straight-forward manner; consequently modern works that depend on it (e.g. Deep Narrow Network) might inherit the blackbox property. Bottom-up constructions for function approximation using NN then emerged, though they also lack the interpretability (see appendix A.3 for more related works). Also consider a demonstration in Nielsen (2015) that could help improve our understanding of universal approximation.

**Outline**. This paper is arranged as the following. Section 2 shows explicit TNN construction, related results, including *a pencil-and-paper example* for pedagogical purpose. Likewise, section 3 shows SQANN construction, statements regarding SQANN, *another pencil-and-paper example*, then experimental results of its application, before we conclude the paper with limitations and future works. Python codes and clearer figures are fully available in supp. materials (also see appendix).

## 2 TRIANGULARLY-CONSTRUCTED NN (TNN)

TNN is the prototype NN for our interpretable universal approximator. SQANN (next section) partially borrows the concept from TNN which will be useful as an easy and manageable illustration to deliver the following ideas: (1) organized activations of neurons and (2) the retrieval of $\alpha$ values as the outputs. The model is $TNN(x) = \alpha^T \sigma(Wx + b)$ where $x \in [0, 1]$, $\alpha, b \in \mathbb{R}^N$ and $W \in \mathbb{R}^N$, where we use sigmoid function $\sigma$ for simplicity. We start with a simple scalar function $y = f(x) \in \mathbb{R}$ for $x \in [0, 1]$, thus TNN's interpretability can be illustrated very clearly.

**Assumption: Linear Ordering**. It is constructed on a linearly ordered dataset containing $N$ samples $\{(x^{(k)}, y^{(k)}) \in \mathbb{R}^n \times \mathbb{R}\}_{k=1}^N$ such that $x^{(N)} < x^{(N-1)} < \cdots < x^{(1)}$ and $y^{(k)} = f(x^{(k)})$, $f$ the true function that TNN will approximate. The interpretability comes from the linear ordering property where higher value of $x$ ($\approx 1$) will activate more neurons while lower values will activate less neurons as shown in fig 1(A). Then $\alpha$ values will be retrieved in a continuous way through dot product, eventually used to compute the output for prediction. In time series, such as ECG (Electrocardiogram), signals can be approximated point-wise (although it is still preferable to have a noise model during preprocessing to prevent overfitting the noise). Meaningful interpretation can be given, for example, by mapping PQRST segments from ECG to specific neurons within TNN, giving some neurons specific meaning and thus interpretability. For more remarks and definition of formal *linear ordering* etc, see appendix A.4.

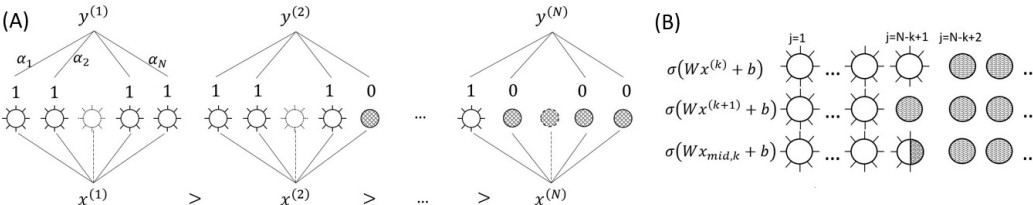

Figure 1: (A) Triangular construction is built by prioritizing interpretability of a neural network. As $x^{(k)}$ decreases in "strength", the neurons are "turned off" correspondingly. (B) Activations of neurons for $x^{(k)}, x^{(k+1)}$ and their mid-point $x_{mid,k}$. Not only will neuron activation be half at the mid-point, the output $y_{mid,k} = \frac{1}{2}(y^{(k)} + y^{(k+1)})$ is also half the sum of its neighbours'.

**Ordered activation**. We would like $x^{(1)}$ to activate all neurons, while $x^{(N)}$ activates only 1 neuron. In other words, ideally $\sigma(Wx^{(1)} + b) = [1, 1, \ldots, 1, 1]^T$, followed by $\sigma(Wx^{(2)} + b) = [1, 1, \ldots, 1, 0]^T$ and so on until $\sigma(Wx^{(N)} + b) = [1, 0, \ldots, 0]^T$; again, refer to fig. 1(A). With this concept, we seek to achieve interpretability by successive activations of neurons depending on the "intensity" of the input, with $x^{(1)}$ being the most intense. In general, the above can be written as

$$\sigma^{(k)} \equiv \sigma(Wx^{(k)} + b) = [\underbrace{1, \ldots, 1}_{N-(k-1)}, \underbrace{0, \ldots, 0}_{k-1}]^T \tag{1}$$

which is approximately achieved for $k = 1, \ldots, N$ at large $a$ (and exactly if $a \to \infty$) with

$$(Wx^{(k)} + b)_j = \begin{cases} \leq -a, j \geq N - k + 2 \\ \geq +a, j \leq N - k + 1 \end{cases} \tag{2}$$

For more remarks, see appendix A.4 subsection *ordered activation*.

**TNN construction: computing weights** $W, b, \alpha$. How then do we compute $W$, $b$ to achieve the *ordered activation*? Consider first $(Wx^{(2)} + b)_N = -a$ and $(Wx^{(1)} + b)_N = a$ and solve them. This yields $W_N = 2a/\Delta^{(1)}$ and $b_N = a - W_N x^{(1)}$ where $\Delta^{(1)} = x^{(1)} - x^{(2)}$. Iterating through k, i.e. solving $(Wx^{(k+1)} + b)_{N-k+1} = -a$ and $(Wx^{(k)} + b)_{N-k+1} = a$ we obtain $W_{N-k+1} = 2a/\Delta^{(k)}$ and $b_{N-k+1} = a - W_{N-k+1}x^{(k)}$ where $\Delta^{(k)} = x^{(k)} - x^{(k+1)}$. We can rewrite the indices so that $W_k = 2a/\Delta^{(N-k+1)}$ and $b_k = a - W_k x^{(N-k+1)}$ whenever convenient. For $\Delta^{(N)}$, we need a dummy $x^{(N+1)}$ value or we can directly choose its value, e.g. $\frac{1}{N}\Sigma_{k=1}^{N-1}\Delta^{(k)}$. The effect is illustrated by the value near $x = 0$ in fig. 2(A1-3) and should not pose any problem; the chosen dummy value will only affect the shape at the left end of the graph.

We compute $\alpha$ using the property of equation (1). From fig. 1(A), this means ideally $y^{(1)} = \Sigma_{i=1}^N \alpha_i \sigma(Wx^{(1)} + b)_i$ for $a \to \infty$, and similarly $y^{(2)} = \Sigma_{i=1}^{N-1} \alpha_i \sigma(Wx^{(2)} + b)_i$ and so on until $y^{(N)} = \alpha_1 \sigma(Wx^{(N)} + b)_1$. Putting them together as $y = [y^{(1)}, \ldots, y^{(N)}]^T$, we get $y = A\alpha$ where $A$ is an upper-left triangular matrix and the inverse $A^{-1}$ exists. Thus, $\alpha = A^{-1}y$, a matrix such that $A_{ij}^{-1} = 1$ along the diagonal, $A_{i,i+1}^{-1} = -1$ and zeroes otherwise, which facilitates a convenient computation. The triangular construction is complete:

$$TNN(x) = \alpha^T \sigma(Wx + b) \tag{3}$$

While Nielsen (2015) provides only visual demonstration, the following result shows rigorous proof on universal approximation at work (python code also available).

**Theorem 1** *TNN achieves arbitrarily high accuracy on the training dataset. Proof: see appendix A.4.1. Also see example results in fig. 2.*

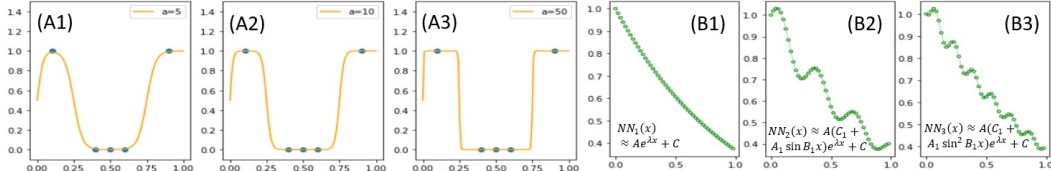

Figure 2: (A1-3) Three triangular constructions (orange plots) using different values of $a$. Higher $a$ results in more step-wise plots and more constant values around the data samples (blue points). (B1) Plots of NNs approximated using triangular construction (smooth green plots) over scatter plots of the corresponding true data (green open circles). The parameters are as the following $A = 1, \lambda = -1, C = 0$ for all, (B2) $A_1 = 0.1, B_1 = 20, C_1 = 1$ (B3) $A_1 = 0.1, B_1 = 10, C_1 = 1$.

With the following proposition, test dataset that resembles training dataset will yield small error. Otherwise, there are out-of-distribution (ood) samples $A \subseteq D'$, which can be incorporated into the training dataset. Catastrophic forgetting is not a problem during re-training because when ood samples from the test dataset are included as training points, each training point $x \in D$ is still identified with a neuron. By theorem 1, $x$ is still approximated accurately. See appendix A.2.2 for remarks on *advantage over existing methods*.

**Proposition 1** *Errors on monotonous interval. Given finite training, test datasets $D, D'$, there exists $A \subseteq D'$ such that, using TNN constructed with $D \cup A$, for all samples in test dataset $(x', y') \in D'$, sample-wise error $e = |y' - TNN(x')|$ has an upper bound $max(|y' - y^{(k+1)}|, |y' - y^{(k)}|)$ for some k. Proof: see appendix A.4.2.*

There is also a mid-point property that can be exploited for generalizability to arbitrarily high accuracy, where data must be sampled such that any instance $x_{test}$ lies inside either (1) the training dataset or (2) is equal to some mid-point of two neighbouring training samples; see the proposition

below. Fig. 1(B) shows how the component of $x_{mid,k}$ at $j = N - k + 1$ is half-activated i.e. the activation value is 0.5. Admittedly, this is an ideal condition for accurate generalizability.

**Proposition 2** *Mid-point property*. *The mid-point* $x_{mid,k} = \frac{1}{2}(x^{(k)} + x^{(k+1)})$ *takes the value of* $\alpha^T \sigma(W x_{mid,k} + b) = \frac{1}{2}(y^{(k)} + y^{(k+1)})$. *Proof: see appendix A.4.3.*

**TNN pencil-and-paper example**. Use TNN to fit the dataset $(x, y) \in \{(1, 1), (0.5, 2), (0, 3)\}$. Then $f(x) \approx TNN(x) = 3\sigma(20x + 5) - \sigma(20x - 5) - \sigma(20x - 15)$. See appendix A.4.4.

Remarks smoothness property, special case, scalability/complexity and generalizability to higher dimensions can be found in appendix A.4.5. We proceed with SQANN, a universal approximation inspired by TNN that allows multi-dimensional input, multi-layer stacking of neurons based on relative strength of neuron activations.

## 3 SEMI-QUANTIZED ACTIVATION NEURAL NETWORK (SQANN)

SQANN architecture is a multi-dimensional universal approximator which (1) retains TNN's idea of using an organized sequence of activations to retrieve $\alpha$, (2) remotely resembles a Radial Basis Function, but (3) has deep neural network properties, such as the possibility of deep learning (multiple constructed layers) and neuron activations. The difference is, a neuron in SQANN corresponds exactly to a data sample as SQANN stores its "fingerprints" as neurons' nuclei as shown in fig. 3(B). The neurons can be activated with different kinds of responses: (1) distinct peaks and (2) half-activations and (3) weak/zero activations, made possible by double selective activation $\sigma_{dsa}$. They are illustrated in fig. 3(A) and the sketch of proof of proposition 3 in appendix A.5.4, *intuition behind $\sigma_{dsa}$*.

The aforementioned distinct types of activation are key to SQANN's main result and give the model an interpretability at least in the following sense: samples are highly/moderately/not recognizable if their activation patterns strongly/moderately/weakly resemble the activation patterns of an existing training sample $x$, where identifications are facilitated by the distinct regions of $\sigma_{dsa}$. The design is "semi" quantized since $\sigma_{dsa}$ has approximately "distinct" levels yet remains continuous. Since SQANN incorporates multi-layer structure, it avoids being a non-generalizing model that nearest neighbours methods suffer from; see the remark in (Pedregosa et al. (2011a)).

**Notations**. The order of data sample within the dataset matters, thus we define our own indexing to prevent confusion. Let the finite training set be $\{(x^{(k)}, y^{(k)}) \in X \times Y : k = 1, 2, \ldots, N\}$. We create the SQANN model that predicts $y^{(k)} = SQANN(x^{(k)})$ with provably perfect accuracy and generalizes well to similar test distribution. Subscript indicates layer, $v$ denotes activation values collected in the "synapses", square bracket with subscript denotes vector component so that $[v_2]_4$ is the 4-th vector component of the activation of layer 2. **Layer k** consists of $(N_k, \alpha_k)$, where $N_k = (\eta_k^{<1>}, \eta_k^{<2>}, \ldots, \eta_k^{<n_k>})$ stores fingerprints/patterns, $\alpha_k = (y_k^{<1>}, y_k^{<2>}, \ldots, y_k^{<n_k>})$ stores output values. The angle bracket denotes the index after relabelling. Hence, if k-th data sample before relabelling is $(x^{(k)}, y^{(k)})$, $k = 50$ and it becomes the first node in layer 2, then we write $\eta_2^{<1>} = \eta_2^{(50)}$. **Concatenation**. To denote the addition of the new $k$-th node to the layer $l$, use $\eta_l^{<k>} \leftarrow v$, where $v$ can be for example $v_2^{(m)}$ the activation of the $m$-th data at layer 2. Alternatively, $N_l \rightarrow concat(N_l, \eta_l^{<k>})$. We can speak about layer $k$ using $N_k$ if $\alpha_k$ is not yet involved. However, once concatenation of $N_l$ is decided, always correspondingly concatenate the $\alpha_l$ i.e. $y_l^{<k>} \leftarrow y^{(m)}$. **Selective clustering** of $p_k = (x_k, y_k)$ for $k = 1, 2$ is loosely defined for $x_1, x_2$ that are close to each other such that: if $y_1, y_2$ are similar, then $p_1, p_2$ are clustered together, otherwise two distinct clusters are created; see appendix for formal definition, its effects on interpolation and more remarks.

**Double selective activation**. Given selective activation $\pi(x, a) = \frac{a}{a + x^2}$ and Super Gaussian $s_g(x, a) = exp(-(x/a)^{2n})$, $n = 4$, $r = 0.5$, then the *double selective activation* is (fig. 3(A)):

$$\sigma_{dsa}(x, a_1, a_2, r) = (1 - r) \times \pi(x, a_1) + r \times s_g(x, a_2) \tag{4}$$

**Nodes activation**. Denote the "synapse" or the activation value of node $j$ at layer $k$ by input $v$ as:

$$[v_k]_j = \sigma_{dsa}(||v - \eta_k^{<j>}||) \tag{5}$$

where $\eta_k^{<j>} \in N_k$ for $j = 1, \ldots, n_k$ and $n_k$ is the number of neurons/nodes in the layer. See fig. 3(B). In SQANN, activations will be forwarded layer by layer, i.e. $[v_1]_j = \sigma_{dsa}\big(||x - \eta_1^{<j>}||\big)$ where $\eta_1^{<j>} \in N_1$ and $[v_{k+1}]_j = \sigma_{dsa}\big(||v_k - \eta_{k+1}^{<j>}||\big)$ where $v_{k+1}^{<j>} \in N_{k+1}$.

## 3.1 SQANN CONSTRUCTION

**Outline of SQANN construction with interpretations**. SQANN is constructed without optimization like gradient descent. Each indexed training data sample is converted into a "fingerprint" or pattern of neuron activations, which undergoes one of the following:

1. *Admission to layer $k$*. Sample's new/distinct fingerprint is added into layer $k$ if the sample weakly activates existing nodes in the layer ($\forall j, [v_k]_j < \tau_{ad}$) and no collision occurs; see fig. 3(D.1).

2. *Collision*. A sample activates one or more neurons strongly i.e. $\exists j, k, [v_k]_j > \tau_{act}$. The earliest layer where collision occurs is denoted $l_c$. Such sample is integrated into $l_c$, thus very similar samples are *selectively clustered*. See fig. 3(D.2); the concept is also illustrated in the sketch of proof for proposition 3, appendix A.5.4.

3. *Filtering into deeper layer* occurs when neither of the above occurs (no strong activation, some moderate activations). Such sample has features loosely similar to previously seen samples, but we need to filter them further to distinguish its finer features.

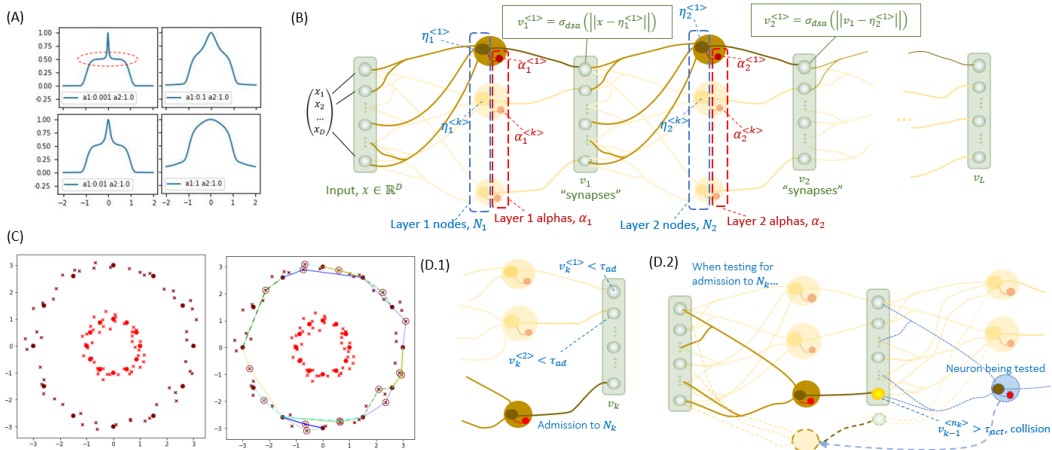

Figure 3: (A) Double selective activation with different parameters. (B) SQANN schematic. Each layer $(N_k, \alpha_k)$ is stylized as a collection of neurons. A neuron stores the main "fingerprint" in nucleus $\eta_l^{<k>}$ (dark brown) and its corresponding "output" in nucleus $\alpha_l^{<k>}$ (dark red). When strong activation is detected, the signal will be redirected to the dark red nucleus $\alpha_l^{<k>}$. (C) SQANN used for a simple classification. (Left) The large filled dots are training samples, x marks are test samples. Bright red indicates $y = 1.0$, dark red $y = 0.5$. (Right) Same as left but test samples that are interpolated (i.e. no strong activation) are annotated with red open circles; colored lines indicate which two training samples are used for the interpolation. Lines are marked with different colors and styles for clarity. (D) Construction of SQANN when (D.1) admission occurs: a new neuron is introduced, creating more connection analogous to mammalian brains. (D.2) collision occurs.

**Layer 1 construction**. To initialize, let $N_1 = (\eta_1^{<1>})$ and $\alpha_1 = (y_1^{<1>})$ where $\eta_1^{<1>} \leftarrow x^{(1)}$ and $y_1^{<1>} \leftarrow y^{(1)}$. Let $\tau_{ad}, \tau_{act}$ be the *admission threshold* and *activation threshold* respectively. We typically set $\tau_{ad} = 0.1, \tau_{act} = 0.9$. We extend the layer to tuples $N_1 = (\eta_1^{<1>}, \ldots, \eta_1^{<n_1>})$ and $\alpha_1 = (y_1^{<1>}, \ldots, y_1^{<n_1>})$ through sample-collection function in the pseudo code 1. To do this, take a new sample $(x^{(k)}, y^{(k)}), k > 2$ and we *check $N_1$ activation*, i.e. let $v_1^{(k)}$ be the activation of current layer by this new sample, i.e. $[v_1^{(k)}]_j = \sigma_{dsa}\big(||x^{(k)} - \eta_1^{<j>}||\big)$ for all $\eta_1^{<j>} \in N_1$. Then, either: (1) new sample is *admitted to $N_1$* as a new distinct node/neuron. If for all $j$ such that $[v_1^{(2)}]_j < \tau_{ad}$, then $N_1 \rightarrow concat(N_1, x^{(k)})$ and $\alpha_1 \rightarrow concat(\alpha_1, y^{(k)})$. (2) *collision* occurs, when there exists $j$ such

that $\tau_{act} < [v_1^{(k)}]_j < 1$, thus sample is admitted via collision resolution mechanism. Exclusively for layer 1, sample will simply be admitted into the layer for selective clustering. New sample causing $[v_1^{(k)}]_j = 1$ collision is unresolvable; see appendix on ill-defined datasets (3) or new sample is *filtered to deeper layers*, when neither occurs. Finally, we complete the iteration over all training data, $N_1 = (\eta_1^{<k>} : k = 1, \ldots, n_1)$ and $\alpha_1 = (y^{<k>} : k = 1, \ldots, n_1)$.

**Lemma 1** *Layer 1 of SQANN achieves arbitrarily high accuracy on training data subset $N_1 \times \alpha_1$.*

Proof: Let $N_1 = (x^{<1>} = x^{(1)}, x^{<2>}, \ldots, x^{<n_1>})$; note that $x^{<k>}$ is not necessarily $x^{(k)}$ except for $k = 1$ for initialization. To prove the lemma, take a sample $(x, y) \in N_1 \times \alpha_1$. Then we must have $x = x^{<j>}, y = \alpha_1^{<j>}$ for some $j = 1, \ldots, n_1$. Since $[v_1]_{j'} = \sigma_{dsa}(||x - \eta_1^{<j'>}||)$ and $\eta_1^{<j'>} = x^{<j'>}$ for all $j' = 1, \ldots, n_1$, we get exactly $[v_1]_j = 1$. Furthermore, for other $i \neq j$, we have $[v_1]_i < 1$ due to the admission conditions (1) and (2) used during *check $N_1$ process*. Finally, computing $y = \alpha_1^{<j>}$ where $j = argmax_{j'}[v_1]_{j'}$, we retrieve the exact value. $\square$

At this point, it may be clearer to readers how SQANN is constructed. In short, for each layer, *representative activations* become the neurons of the layer. In layer 1, representatives activations are the samples themselves. In deeper layers, they are activations propagated to the layer.

**Layer k construction**. Layer $k$ construction is similar to layer 1 construction, except collision could occur at any layer $l_c \leq k$ (next paragraph). Assume every layer $l \in \Lambda = \{1, \ldots, k - 1\}$ have been constructed using $X_{i \in \Lambda} \subseteq X$. Assume there are still unused data samples i.e. $U = X \setminus \{\bigcup_{i=1}^{k-1} X_i\}$ is non-empty, obtained from samples that have been *filtered to deeper layers*. Let $U = \{u^{<1>}, u^{<2>}, \ldots\}$ after re-labelling the indices, with corresponding output values $\{y^{<1>}, \ldots\}$. Initialize by first checking $v_k^{<1>}$, the activation of $u^{<1>}$ at layer $k$ for collision; if collision occurs, see next paragraph, otherwise, set $N_k = (\eta_k^{<1>}), \alpha_k = (y_k^{<1>})$ i.e. $\eta_k^{<1>} \leftarrow v_k^{<1>}$. Similar to layer 1 construction, perform *check $N_k$ activation* on $(u^{}, y^{})$ for $i > 1$ by computing activation $v_k^{}$ and checking it against the existing nodes. One of the three cases occur (1) admission, when $[v_k^{}]_j < \tau_{ad}$ for all $j = 1, \ldots, n_k$ and no collision (2) collision, when there exists index $j$ at a *collided layer* $l_c \leq k$ such that $[v_{l_c}^{}]_j > \tau_{act}$ or (3) otherwise. If (1) occurs, the activation $v_k^{}$ is added as a new neuron to the layer, $N_k \rightarrow concat(N_k, v_k^{})$. If (3) occurs, filter the data sample for deeper layer. Assuming no collision, the process is repeated for the next unused data sample $u^{}$ until all remaining data samples are checked. Once done, repeat the process for layer $k + 1$ construction.

Suppose collision happens when we check $u^{<m>}$ at layer $l_c$, we use the collision resolution mechanism. We destroy all layers $l > l_c$ and put the collided sample into $l_c$, i.e. $N_{l_c} \rightarrow concat(N_{l_c}, v_k^{<m>})$ and $\alpha_{l_c} \rightarrow concat(\alpha_{l_c}, y_k^{<m>})$ (push-node in the pseudo code). No layer will be destroyed if $l_c = k$, the current layer. The data samples used in each destroyed layer are returned to the list of unused samples in the same order they have been used during the construction (return-index() in the pseudo code); we refer to this as *order integrity*. Once the colliding sample is added to $l_c$, effectively, the strong activation in this layer is now overshadowed by maximum activation (*selective clustering* in action), since the exact neuron is now included as the representative of itself and its locality. This is possibly a practically inefficient process, since we tear down intermediate layers, but we only prioritize the completion of the construction for now.

**Computing output via SQANN propagation (prediction)**. Let an input be $x$. The output $y = SQANN(x)$ is computed by propagating and processing signals through the layers; fig. 3(B). Then $[v_1]_j = \sigma_{dsa}(||x - \eta_1^{<j>}||)$. If there exists $j$ such that $v_1^{<j>} > \tau_{act}$, then set $y = \alpha_1^{<j>}$ where $j = argmax_{j'}[v_1]_{j'}$. Otherwise, for subsequent layer $k$, recursively compute $[v_k]_j = \sigma_{dsa}(||v_{k-1} - \eta_k^{<j>}||)$ for all $j = 1, \ldots, n_k$. If there exist $j, k$ such that $[v_k]_j > \tau_{act}$, then $y = \alpha_k^{<j>}$ where $j = argmax_{j'}[v_k]_{j'}$. If such layer $k$ is not found, we have to perform interpolation.

Interpolations can be done in many different ways, and this paper implements a simple interpolation using values from the two most strongly activated neurons. Suppose $V_1 = [v_m]_i$ and $V_2 = [v_n]_j$ are the two most activated neurons, then the interpolated value can be, for example, $y = \frac{V_1[\alpha_m]_i + V_2[\alpha_n]_j}{V_1 + V_2}$. The form of interpolation can be adjusted according to the knowledge we

have on the dataset, e.g. we can use TNN with high $a$ if we know the dataset is locally constant. See appendix A.5.2 for illustration and remarks.

Each training sample $x$ admitted to layer $N_k$ leaves a fingerprint, in the sense that it has a collection of activations $\{v_l | l = 1, \ldots, k\}$ as it is SQANN-propagated through the neural network. This collection is unique amongst training samples, especially because of $[v_k]_j = 1$ where $j$ is an index within layer $k$ it is admitted into. Furthermore, locality is preserved to the extent that if $x' \approx x$, then activation $[v'_k]_j \approx 1$ is around the peak and thus, by SQANN propagation, due to argmax, it is likely we retrieve $y$, the ground-truth value corresponding to $x$. For now, we only use argmax, but more subtle adjustment can be done to obtain $y' \approx y$ but $y' \neq y$ for $x'$ by incorporating information about the manifold at that locality, if such knowledge is available. See appendix A.5.3 on *good practice for scalability*.

**Completing construction**. Due to collisions, readers might wonder if the construction will complete at all. During collision, layers are torn down and reconstructed. Suppose during layer $k$ construction, collision occurs at layer $c$ for $c < k$. Upon reconstruction back to layer $k$, layer $c$ may be torn down again in the next collisions. Is it possible that collision occurs infinitely cyclically? The following proposition addresses the concern through *order integrity* previously mentioned.

**Proposition 3** *SQANN construction completes with high probability $p \approx 1$. See appendix A.5.4 for (1) sketch of proof and a required assumption (2) stronger assumption needed for $p = 1$.*

**Arbitrarily high accuracy** on training dataset $D$ is relatively simple to prove in the following theorem: roughly for each $(x, y) \in D$, there exist $l, k$ such that $x$ maximally activates the node $\eta_l^{<k>}$, thus the correct $y$ is guaranteed to be fetched from $\alpha_l$. Catastrophic forgetting resistance is proven similarly: when new samples are used for training, previous samples are not forgotten since SQANN stores the particular fingerprint $\eta_l^{<k>}$ for each sample.

**Theorem 2** *Assume SQANN construction is completed. SQANN achieves arbitrarily high accuracy on a training dataset. Furthermore, it is resistant to catastrophic forgetting. Proof: see appendix A.5.5. Note: Our code provides experimental demonstrations showing zero errors on all training samples; see example results in fig. 3(C).*

**SQANN pencil-and-paper example**. With $a_1, a_2 = 0.001, 0.5$, $\tau_{ad}, \tau_{act} = 0.1, 0.9$, create SQANN universal approximator for indexed data $X = [x^{(1)}, x^{(2)}, x^{(3)}, x^{(4)}] = \begin{bmatrix} 1 & 1.2 & -1 & -1.2 \\ 1.2 & 0.8 & -1 & -1.2 \end{bmatrix}$ and $Y = [y_1, y_2, y_3, y_4] = [1, 1, 0, 0]$. See appendix A.5.6 for more questions and solutions.

## 3.2 EXPERIMENT TO TEST GENERALIZABILITY OF SQANN

**Test datasets with increasing spread from training distribution**. The accuracy of SQANN on high-dimensional dataset outside the training dataset is harder to formalize in theorems. Furthermore, real life data is often noisy and possibly not regularly structured. We avoid making any related statements for SQANN for now. We instead provide empirical results on test datasets that are *similar* to the training dataset, to the extent that each point in the test dataset is a training sample perturbed by uniform random values of increasing magnitude. We refer to the noise magnitude as the *test data spread*. Fig. 4(A,A.2) show four domains $X$ with different test data spread. Test dataset that has larger test data spread contains data samples that are noisier and further away from the training data points. Fig. 4(B,B.2) show that SQANN naturally performs better with smaller test data spread. As the test data spread increases, larger errors are observed. Likewise, smaller spread means smaller $N_{interp}$, i.e. fewer data samples fail to activate neurons in SQANN strongly. For all, training errors are 0 *as expected from theorem 2*.

**Classification and visualization of SQANN's special interpretability features**. We show the use of SQANN for a simple classification problem in fig. 3(C). The ring outside is labelled 0.5, while the ring inside 1.0. With activation parameters $a_1 = a_2 = 1$, we achieve zero error not only for training dataset (to be expected from theorem 2) but also on test dataset. A special feature in SQANN is its ability to tell the user which data samples fail to activate any neurons strongly; such samples' output must be interpolated (see SQANN propagation). In fig. 3(C) right, points marked with red open circles need interpolation. Each such point is interpolated using two training samples whose "fingerprint" neurons are most strongly excited. These two samples are shown as the two

points directly linked by colored straight lines to the open circle. This is possible because SQANN systematically stores indices of training data samples within respective layers. The list of indices organized by layers can even be explicitly printed e.g. see SQANN.ipynb, supp. materials.

**SQANN is tested for regression on Boston Housing and Diabetes Datasets** to demonstrate its generalizability to unseen/test samples, as the following (simplified here): (1) A small subset of samples $D \subseteq \mathcal{D}$ (the first 20% of full dataset $\mathcal{D}$) is used to train SQANN and 9 other regression methods. (2) Mean Squared Errors (MSE) values are measured on unused data $D_{test}$ on all 10 models; we expect large errors on some test samples. (3) SQANN's activations are used to collect samples with large absolute errors $e_\tau(x) = |SQANN(x) - y_0| > \tau$ and we treat them as out-of-distribution (OOD) samples. These samples are considered as new distinct samples to be integrated into $D$ as the new training dataset $D'$. (4) Train the 10 models, now with $D'$. (5) Then MSE is measured again on $D_{test}$ (yes, there will be partial overfitting). From Boston dataset: for $\tau = 5$, SQANN MSE improved from 9.90 to 3.08. Decision tree improves the most with $D'$ (7.36 to 2.07); see table 1. For more details and diabetes dataset, see the appendix A.5.7.

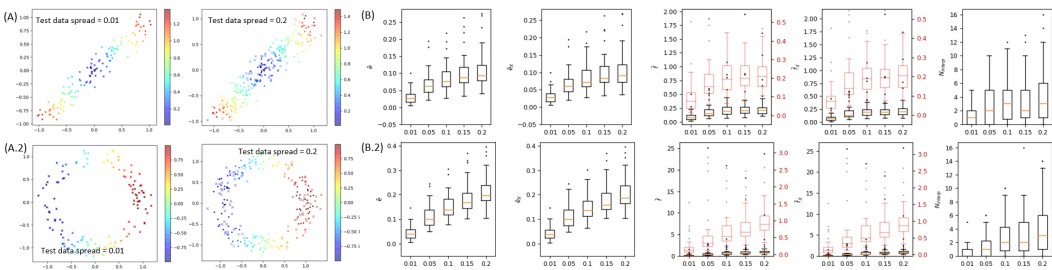

Figure 4: (A) Training/test (circles/x marks) data for demonstration. Smaller/larger test data spread means test samples are closer/further to/from training samples. (B) Boxplots for data whose distributions are similar to (A). Column 1(3): (fractional) errors on test data samples. Column 2(4), (fractional) errors on test data samples excluding interpolated samples. Column 5: no. of data samples whose predicted values are interpolated. (B.2) Similar to top, but for (A.2).

## 4 CONCLUSION, LIMITATION AND FUTURE DIRECTION

**Limitations and future directions**. **TNN** is clearly limited in regards to its application to multi-dimensional input data although it can be useful on different types of time series data. **SQANN** limitation and possible future development currently include 1) simple sequential drawing of samples that may result in the imbalance of layer size. In the future, more sophisticated ordering of training samples can be used so that layers are constructed with meaningful and purposeful arrangement e.g. deeper layers can be purposefully reserved for rare cases; more research on this is necessary to optimize the results 2) layer destruction during the treatment of collision cases might be an inefficient mechanism, which can be improved in the future.

**Conclusion**, we have proposed TNN and SQANN, two interpretable NNs for universal approximation designed to (1) be resistant to catastrophic forgetting (2) have provably high accuracy on training datasets and (3) can be used to handle out-of-distribution samples.

**Buffer page**. Just in case.

Table 1: Comparing MSE on different regression methods for Boston Housing dataset. Row o., or original, shows MSE obtained from models trained on $D$. Row eT shows MSE from models trained on $D'$ with $\tau = T$. All are evaluated on $D_{test}$.

|    | Lin   | Ridge | Lasso | LSVR  | NuSVR | SVR   | DTree | kneigh | MLP   | SQANN |
|----|-------|-------|-------|-------|-------|-------|-------|--------|-------|-------|
| o. | 36.45 | 7.990 | 9.834 | 8.355 | 8.833 | 8.712 | 7.356 | 7.393  | 12.77 | 9.898 |
| e5 | 5.139 | 5.135 | 7.068 | 5.950 | 6.295 | 6.068 | 5.026 | 4.798  | 3.481 | 7.998 |
| e2 | 4.993 | 5.028 | 7.882 | 5.832 | 6.121 | 5.895 | **2.072** | 3.025  | 3.846 | **3.076** |

**Note on table 1**. The entries in the header denote the models available in scikit-learn (Pedregosa et al. (2011b)): Lin, Ridge and Lasso are the linear models: linear , Ridge (linear least square with L2 regularization), Lasso (linear with L1) respectively; the Support Vector Regression models: LinSVR, NuSVR, SVR are respectively linear SVR, $\nu$-SVR and $\epsilon$-SVR. DTree: Decision tree; kneigh: k neighbours are selected from the best $k = 2, 3, ..., 16$, MLP: multi-layer perceptrons, or the fully-connected neural network, with 2 layers, each layer having 64 neurons each trained for a max of 12000 iterations (convergence is attained for both). For SQANN, the initial model trained on $D$ is kept after SQANN' is trained on $D'$. Thus, we can choose results based on the strength of activations between SQANN and SQANN'.

For Boston Housing Dataset $\tau = 5$ (i.e. e5 of table 1), using SQANN we integrated 211 samples from the test dataset into training dataset, so $|D'| = 311$. Overall, $0.615$ of the whole $\mathcal{D}$ is used for new training. For $\tau = 2$, i.e. e2 of table 1, using SQANN we integrated 319 samples, so $|D'| = 419$ i.e. $0.828$ of the whole $\mathcal{D}$ is used for new training. With $\tau = 2$, regression performance of SQANN improves greatly compared to other methods, except for decision tree regression. We have thus also seen that SQANN can be used to perform sample selections for data that appear to be out of distribution; this has improved decision tree performance greatly. The performance of other models have improved reasonably too, especially MLP. For MLP, however, the randomness used to achieve convergence to some local minima might have led it to explore other minima; hence we get slightly decreased performance for e2 compared to e5.

```
def fit_data(X,Y):
  # Main SQANN loop
  l_now=1 # layer now
  while True:
    ssig, collision = sample_collection(X, Y, l_now)
    if ssig is 'no_more_data':
      break
    elif ssig is 'collision':
      l_c = collision['collided_layer']
      for l_j from l_c+1 to l_now+1:
        return_index(l_j)
      kp = collision['perpetrator_index']
      push_node(kp,X[kp,:],Y[kp],l_c)
      l_now = l_c
    l_now+=1
```

```
def sample_collection(X,Y,layer):
  i=unused_indices[0]
  x=X[i,:]
  x, collision = forward_cons(x,layer−1)
  ssig, collision = check_signal(collision)
  nodes, node_values = new_nodes(x,Y[i])
  remove_index(i,layer)
  for i in unused_indices:
    x=X[i,:]
    x, collision = forward_cons(x,layer−1)
    ssig, collision = check_signal(collision)
    act=activate(x,nodes)
    if all(act<admission_threshold)
      update_nodes(x,Y[i],nodes,node_values)
      remove_index(i,layer)
  return ssig, collision
```

Pseudo code 1: Pseudo code for the construction of SQANN. The function *activate* corresponds to equation 5. See appendix B.2 for mapping to python code.

## ETHICS STATEMENTS

This paper introduces function approximators with novel properties. It is purely mathematical and algorithmic. No specific ethical issues are present. The ethical context depends only on the dataset but in this paper, only common public datasets have been used.

## REPRODUCIBILITY STATEMENTS

All codes are available in the supp. materials (to be released to public repository in case of acceptance). Results are easily reproducible even with without random number seeding since data distributions are sufficiently controlled. Jupyter notebook for our particular results are also present. Proofs are all included in the appendix, with sketch of proof and additional statement of assumptions where applicable.

## ACKNOWLEDGMENTS

Anonymous for now.

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

# A APPENDIX

We have reorganized the content based on ICLR 2022 reviews. Also, contents that are less relevant are moved to the Extra section B at the end of appendix.

## A.1 ALGORITHM CODE AND SUPPLEMENTARY MATERIALS

All the codes are available in the supplementary material, and the link to the repository will be made public if the paper is accepted. README.md lists the necessary commands to obtain the results found in this paper. Results in Jupyter notebooks format are also included.

## A.2 CONCEPT DISAMBIGUATION

Link back to main text introduction, section 1.

### A.2.1 UNIVERSAL APPROXIMATION

Universal approximation by TNN and SQANN is not restricted to specific conditions. Traditional approximation may require strong conditions e.g. Cybenko (1989) shows result on $C(I_n)$, space of continuous function. Continuous function that approximates two points necessitate continuous interpolation for any point along the path between these two points. However, hypothetically, pathological set of data points may exist where the value is not even continuous along the path. Our SQANN, for example, can handle this since we allow user to choose the interpolation function in the case interpolations are opted for output computation (e.g. during weak activations).

### A.2.2 CATASTROPHIC FORGETTING

**Catastrophic forgetting and online learning**. Continual or online learning have posed some known challenges. In Kirkpatrick et al. (2017), catastrophic forgetting is defined as "the tendency for knowledge of previously learnt task(s) (e.g. task A) to be abruptly lost as information relevant to the current task (e.g. task B) is incorporated". In our context, we can distinguish task A from task B as old and new dataset. Learning from the new dataset which has possibly different distribution from the old dataset might degrade the performance of the model.

**Analogy to biological system and resistance to catastrophic forgetting**. Unlike artificial NN, mammalian brain retains old information when it learns new information *by protecting previously acquired knowledge in neocortical circuits*; see Kirkpatrick et al. (2017) and the references thereof. As rats learn new skills, the volumes of dendritic spines in their brains increase (Yang et al. (2009)) while existing dendrites persist, thus they retain old memories. Both TNN and SQANN have similar property. Particularly, SQANN increases the size of a layer as it progressively acquires new samples during construction (learning). This inevitably increases the number of weights that connect the layers; see fig. 3(D.1, D.2).

Furthermore, they exhibit *resistance to catastrophic forgetting*. In TNN, this is simply because the old sample $x_{old}$ still yields activation with signature $[1, 1, ..., 1, 0, 0, ..., 0]$, in which the last activated neuron (taking the value 1) is still identified with the old sample. For SQANN, each old sample is not forgotten since its exact "fingerprint" is already registered (i.e. input $x_{old}$ has been converted) to a neuron's nucleus $\eta_l^{<k>}$ for some $l, k$, as shown in fig. 3(B). The activation pattern of a sample $x_{old}$ includes the value 1 in a specific node of a "synapse", $v_l^{<k>}$ and the particular combinations of values in other synapses.

**Advantage over existing methods**. Existing ML methods do not typically admit out-of-distributions easily; i.e. even if we re-train them on the new ood samples, they may end up with poor prediction on ood. As an illustration, a linear regression that includes ood samples may shift the gradient a little, but ood sample is still far from the regression line. To make the matter worse, when there are many ood enough to change the distribution of the data sample significantly, the model may *forget* the previous distribution: we consider this catastrophic forgetting.

## A.3 RELATED WORKS

Older works on the construction of universal approximators do not focus on interpretability as well, thus readers have to observe for themselves the shape of the resulting networks from the components used in the construction. For example, there are works related to spline functions like Mhaskar & Micchelli (1992); Mhaskar (1993b;a); Chui et al. (1994); another example by Sartori & Antsaklis

(1991) where two-layer NN can be formed by arbitrarily choosing the weights of first layer and computing the weights of second layer explicitly; and section 5 of Pinkus (1999) shows the equations used to obtain weights and proves the existence of such solutions. In some of them and others such as Mhaskar (1996); Chui et al. (1996), the focus lies in error quantification.

## A.4 APPENDIX FOR TNN

Link back to main text section 2.
The traditional definition of NN with a single hidden layer is given by $\Sigma_i \alpha_i \sigma(y_i^T x + b_i)$ following equation (1) from Cybenko (1989), with $x, y_i \in \mathbb{R}^n, b_i \in \mathbb{R}$ where $\sigma$ is any sigmoidal function with the property specified in the paper. A familiar example of a sigmoidal function is the sigmoid function $1/(1 + e^{-x})$. Compared to the traditional version, TNN has a slight generalization on the weights $W$, which is also a form that has been used in modern implementation.

**Assumption: linear ordering**. Linear order is any binary relation with (1) reflexivity (2) transitivity (3) anti-symmetricity (4) $x \leq y$ or $y \leq x$. As is customary (Mhaskar (1993a)), the domain of the function is limited to $[0, 1]^n$ where $n$ is the number of dimensions of the input, justifiable for practical dataset with finite domain easily scaled to $[0, 1]$. Some readers might ask how linear ordering can be performed meaningfully. Also, if linear ordering is done by human, interpretability is not improved, since human user already knows about ordering. To answer that, we again refer to the ECG example we gave earlier: time series is a naturally linearly ordered data. Next, linear ordering in TNN can be seen as a parameter strength tuning, useful for example when TNN is incorporated in a larger system.

**Ordered activation**. Recall that we would like $x^{(1)}$ to activate all neurons, while $x^{(N)}$ activates only 1 neuron, so that eventually we achieve something like fig. 1(A). To approximately fulfil the *ordered activation* conditions stated in the main text, we first need $(Wx^{(1)} + b)_j \geq a$ for all $j$, where sub-script $j$ denotes the $j$-th component in the vector, to distinguish from superscript $(k)$ which denotes the $k$-th data sample according to the linear ordering. The next iteration will be $(Wx^{(2)} + b)_j \leq -a$ for $j = N$ and $(Wx^{(2)} + b)_j \geq a$ for $j = 1, \ldots, N - 1$, and similarly for other $x^{(k)}$ for $k = 3, \ldots, N$. With this, we attain eq. (1) and (2).

Since we use sigmoid function, these conditions are ideal and not strictly attainable, because sigmoid function asymptotically achieves 0 and 1 at infinities. Nevertheless, we show later that we can achieve arbitrarily small error $\epsilon$ by adjusting *activation threshold* $a$. We define the activation threshold to be the value $a$ such that $\sigma(a) = 1 - \delta$, where $\delta$ is a small number. For this paper, fixing $a = 5$ is sufficient. The shape of sigmoid function is convenient enough to be symmetrical in the sense that $\sigma(-a) = \delta$ since $1/(1 + e^{-a}) = 1 - \delta$ can be rearranged to $1/(1 + e^{-(-a)}) = \delta$, useful for the proof later.

### A.4.1 TNN THEOREM PROOF

**Theorem 1**. TNN achieves arbitrarily high accuracy on the training dataset.
Proof: We need equations (1) and (2). In practical situation, where $a$ is finite, we therefore have $\sigma_j^{(k)} \geq 1 - \delta$ for $j \leq N - (k - 1)$ and $\sigma_j^{(k)} \leq \delta$ for $j > N - (k - 1)$ and $\delta > 0$. Then, define sample error as $e^{(k)} = |y^{(k)} - [A^{-1}y]^T \sigma^{(k)})|$ and average error per sample will be $e = \frac{1}{N}\Sigma_{i=1}^N e^{(k)}$. Then $e^{(k)} \leq \delta(N + 1)U$ where $U = \max_k |y^{(k)}|$ is the upper bound for the absolute value of the function over all samples (see proof below). Hence, setting $\delta = \frac{\epsilon}{U(N+1)}$ guarantees that $e \leq \epsilon$. Since $\delta$ can be monotonously decreased by increasing $a$, we have shown that arbitrarily small error $\epsilon$ can be achieved in this approximation; see fig. 2(B1-3) for plotted examples. Note that $e^{(k)} = 0$ iff $\delta = 0$ iff $a = \infty$.
Show that $e^{(k)} \leq \delta(N + 1)U$ where $U = \max_k |y^{(k)}|$. We abbreviate $\sigma^{(k)}$ as $\sigma$, fixing k.

$$e^{(k)} = |y^{(k)} - (A^{-1}y)^T \sigma^{(k)}| = \left|y^{(k)} - \left(y^{(N)}\sigma_1 + (y^{(N-1)} - y^{(N)})\sigma_2 + \cdots \right.\right.$$
$$\left. + (y^{(k+1)} - y^{(k+2)})\sigma_{N-k} \quad\right\} \sigma_j \geq 1 - \delta$$
$$+ (y^{(k)} - y^{(k+1)})\sigma_{N-k+1}$$
$$+ (y^{(k-1)} - y^{(k)})\sigma_{N-k+2}$$
$$\left. + (y^{(k-2)} - y^{(k-1)})\sigma_{N-k+3} + \cdots \right\} \sigma_j \leq \delta$$
$$\left.+ (y^{(1)} - y^{(2)})\sigma_N \right)\Bigg|$$

which we can rearrange according to $y^{(j)}$ instead to

$$e^{(k)} = \left|y^{(k)} - \left(\Sigma_{j=k+1}^N y^{(j)}[\sigma_{N-j+1} - \sigma_{N-j+2}]\right.\right.$$
$$\left.\left. + y^{(k)}[\sigma_{N-k+1} - \sigma_{N-k+2}] + \Sigma_{j=2}^{k-1} y^{(j)}[\sigma_{N-j+1} - \sigma_{N-j+2}] + y^{(1)}\sigma_N \right)\right|$$

Rewriting $d_i = \sigma_{i+1} - \sigma_i$, we now have

$$e^{(k)} = \left|\Sigma_{j=k+1}^N y^{(j)}d_{N-j+1} + y^{(k)}[1 - (1 - \delta - \delta)] + \Sigma_{j=2}^{k-1} y^{(j)}d_{N-j+1} + y^{(1)}\sigma_N \right)\right|$$

As either $\sigma_i \geq 1 - \delta$ or $\sigma_i \leq \delta$, we have $|d_i| \leq \delta$ for all applicable $i$, and using triangle inequality,

$$e^{(k)} \leq |y^{(k)}|\delta + \Sigma_{j=1}^N |y^{(j)}|\delta \leq \delta(N+1) \max_k |y^{(k)}|$$

and we are done. $\square$

### A.4.2 TNN ERROR BOUND AND RESISTANCE TO CATASTROPHIC FORGETTING

**Proposition 1. Errors on monotonous interval.** Given finite training, test datasets $D, D'$, there exists $A \subseteq D'$ such that, using TNN constructed with $D \cup A$, for all samples in test dataset $(x', y') \in D'$, sample-wise error $e = |y' - TNN(x')|$ has an upper bound $max(|y' - y^{(k+1)}|, |y' - y^{(k)}|)$ for some $k$.

Proof: Trivial cases occur for example when (1) most samples in test datasets are novel or out of training dataset distribution. The proposition will hold true trivially because then $A = D'$ i.e. all test data will be fit into TNN. (2) Most samples are already within the distribution, so sample-wise error $e$ is already within bound.
Consider non-trivial case. Since sigmoid function is monotonous, then for any $x \in [x_k, x_{k+1}]$, we have $TNN(x) \in [y_k, y_{k+1}]$. The error of a test sample is $e = |y' - TNN(x)|$. If $x' \in [x_k, x_{k+1}]$ and $y' \in [y_k, y_{k+1}]$, the sample-wise error upper-bound result already holds. We say that such sample is approximately within the training dataset distribution. Note that if $e$ turns out to be too large, i.e. the gradient within the interval is too large, we can always include the test data sample in the training dataset to make a more regular model.

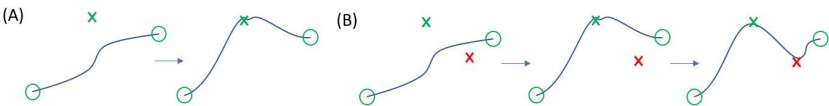

Figure 5: Green circles: training data samples used in TNN. Green and red x: test data samples. (A) The inclusion novel or unseen test sample into the training dataset (B) The inclusion of one test sample (green x) causes another test sample (red x) to be out of distribution, and thus it needs to be included also.

Otherwise, if $y' \notin [y_k, y_{k+1}]$, then we say the sample is out of training distribution because monotonicity of sigmoid function prevents the model from possibly making the correct prediction. Sample-wise error upper-bound no longer holds. By including this sample to the training dataset and recomputing the weights, we recover the upper-bound for this sample. Any such sample is then

included as the family of subset $A \subseteq D'$. For each test data sample, check whether $y' \in [y_k, y_{k+1}]$ for some $k$ and include them into $A$ if it is out of training distribution. This is shown in fig. 5(A). However, the remaining test dataset $U = D' \setminus A$ does not automatically fall nicely into "within training distribution" category. This is best illustrated in fig. 5(B). However, by repeatedly running the same procedure to $U$ and expanding $A$, we will eventually terminate at a point where all $(x', y') \in U$ is within the expanded training distribution $D \cup A$, or reaches the first trivial case. $\square$

As seen above, TNN is resistant to catastrophic forgetting for clear reasons: each data sample in $D$ still corresponds to one exact neuron in TNN, even after adding $A \subseteq D'$.

### A.4.3 TNN MID POINT

**Proposition 2. Mid-point property**. The mid-point $x_{mid,k} = \frac{1}{2}(x^{(k)} + x^{(k+1)})$ takes the value of $\alpha^T \sigma(W x_{mid,k} + b) = \frac{1}{2}(y^{(k)} + y^{(k+1)})$.

Proof: $\sigma_j^{mid} \equiv \sigma(W x_{mid,k} + b)_j = \sigma(\frac{1}{2}(W x^{(k)} + b) + \frac{1}{2}(W x^{(k+1)} + b))_j$. Then $\sigma_{N-k+1}^{mid} = \sigma(\frac{1}{2}a + \frac{1}{2}(-a)) = \sigma(0) = 0.5$. For $j \leq N - k$, $\sigma_j^{mid} = 1$, while for $j \geq N - k + 2$, $\sigma_j^{mid} = 0$. The resulting output of the neural network looks like $\alpha^T[\ldots, 1, 0.5, 0, \ldots]^T$, which is equal to $\frac{1}{2}\alpha^T[\ldots, 1, 1, 0, \ldots]^T + \frac{1}{2}\alpha^T[\ldots, 1, 0, 0, \ldots]^T = \frac{1}{2}(y^{(k)} + y^{(k+1)})$ $\square$.

### A.4.4 TNN EXAMPLE

**TNN pencil-and-paper example**. Use TNN to fit the dataset $(x, y) \in \{(1, 1), (0.5, 2), (0, 3)\}$. Then $f(x) = 3\sigma(20x + 5) - \sigma(20x - 5) - \sigma(20x - 15)$. Suppose $a = 5$ and we have a dataset $\{(1, 1), (0.5, 2), (0, 3)\}$ so that $N = 3$ and $x^{(1)} = 1 > x^{(2)} = 0.5 > x^{(3)} = 0$, which is evenly spaced thus we can use the simplified formula, for example, $W_k = 2 \times 5 \times (3 - 1) = 20$ for $k = 1, 2, 3$ and $b_1 = 5(3 - 2 \times 1)$ etc. Then $W = [20, 20, 20]^T$, $b = [5, -5, -15]^T$ and

$$\alpha = A^{-1}y = \begin{pmatrix} 0 & 0 & 1 \\ 0 & 1 & -1 \\ 1 & -1 & 0 \end{pmatrix} \begin{bmatrix} 1 \\ 2 \\ 3 \end{bmatrix} = \begin{bmatrix} 3 \\ -1 \\ -1 \end{bmatrix}$$

$$f(x) = \alpha^T \sigma(Wx + b) = [3, -1, -1]\sigma([20, 20, 20]^T x + [5, -5, -15]^T)$$

where $\sigma$ is applied component-wise, thus $f(x) = 3\sigma(20x + 5) - \sigma(20x - 5) - \sigma(20x - 15)$.

### A.4.5 MORE REMARKS

**Smoothness**. From the construction, assuming sigmoid function as the activation function, it is obvious that the function is continuous for finite $a$. As $a$ increases, the function becomes more and more constant around each data sample as shown in fig. 2(A1-3), i.e. becoming more step-wise.

**Special case**. When the dataset is evenly spaced, $x^{(k)} = 1 - (k-1)\Delta$, $k = 1, ..., N$, $\Delta = 1/(N-1)$, the results simplify to $W_k = 2a(N - 1)$ and $b_k = a(3 - 2k)$. Not only equation (2) is fulfilled, we also get $(W x^{(k)} + b)_j = a(1 + 2[N - k + 1 - j])$. For the k-th data sample, the activation will then be well-spaced in an interval of $2a$, so that $\sigma^{(k)} = \sigma([\ldots, -3a, -a, a, 3a, \ldots])^T \approx [\ldots, 0, 0, 1, 1, \ldots]^T$.

**Scalability and complexity**. The bulk of memory space usage comes from $W \in \mathbb{R}^{N \times n}$. $N$ is the number of available data points, and this is an unusual feature compared to modern DNN architecture. Since $N$ in common datasets can grow very large, the space complexity becomes $\Omega(Nn + mn)$, still linear w.r.t $N$. Since it is more likely that $m < N$, it is reasonable to simplify to just $\Omega(Nn)$. To reduce the complexity, we can pick a set of representative data points $X_{rep} = \{x_{rep}\}$ for NN construction. The selection of representatives depends on our error tolerance, and it can be done through other machine learning methods, such as clustering. Other data points can then be used for validation. The resulting complexity $\Omega(N_{rep}n)$ will thus highly depend on the variability and the structure of the dataset where $N_{rep} = |X_{rep}|$. Time complexity is almost irrelevant for now, since we are not able to find any meaningful way to compare with the training process of modern DNN. As far as we know, there is no decisive rule on how many epochs are necessary for a DNN training through back-propagation.

**Generalizability to n-dimensional output**. Generalization to scalar input and multi-dimensional output is relatively simple. From equation (3), we can treat $\alpha$ as the coefficients for the only

component of one-dimensional $y$. Generalizing to $y \in \mathbb{R}^m, m > 1$, identify each vector $\alpha_i$ with the component $y_i$. Stacking them up, we can redefine $\alpha = [\alpha_1^T; \alpha_2^T; \dots]$ where a semi-colon denotes the next row, and the construction is done. Note that now $\alpha \in \mathbb{R}^{m \times N}$.

Further generalization to n-dimensional input has been unsatisfactory (see later section of appendix, B.1). Instead, SQANN has been developed with additional mechanism that TNN does not possess.

## A.5 APPENDIX FOR SQANN

Link back to main text section 3.

### A.5.1 SELECTIVE CLUSTERING

**Selective clustering** has been loosely defined in the main text for reasons that will be clear after this. Formally and *more generally*, selective clustering is defined as the following. Let $A = X \times Y$ be a set, and $(x, y)$ be a point. Define $d(A, x)$ as the minimum distance between $x$ and all points in $X$, $x_1 = argmin_{x' \in X} |x - x'|$; let $(x_1, y_1) \in A$. Given error tolerances $\delta, \epsilon > 0$. Let $d(X, x) \leq \delta$. If *output values are similar* (first case), $|y - y_1| \leq \epsilon$, then add $(x, y)$ as a new member of $A$. Otherwise (second case), if *output values are distinct*, $|y - y_1| > \epsilon$, let $(x, y)$ form its own cluster. In short, a new cluster is created (at least nominally) if we have neighbours with distinct $y$ values; they are neighbours, but we may be looking at points at different sides of classification boundaries. This can be easily generalized such that $(x, y)$ is added to another set $B$ where $|y - y_2| < \epsilon$ for $(x_2, y_2) \in B$.

As mentioned in the main text, this does have a role for interpolation during SQANN propagation. If $y, y_1$ are similar, then approximation is likely good. Suppose $x_z$ is near both $x$ and $X$. In the first case (output values are similar), taking interpolation between strongly activated neurons within $X$ (including $x$) will yield approximately the same value as taking argmax (the most strongly activated neuron). This is the nice case. However, in the second case (output values are distinct), we can get unstable result. This is because argmax might tilt between the two different values in, while interpolation could yield the "averaged" value which may not reside in either cluster. The concept selective clustering is thus introduced with the express purpose of talking about such situation. During collision, if two samples strongly activate the neurons but have distinct values, we must be careful about using argmax or deciding to select our interpolation methods. In this paper, only argmax is used; other variations will be left for further studies. For example, when $x_z$ causes activation of 0.99 on one neuron and 0.92 on another. Both are strong activations, but with argmax-based selective clustering, $x_z$ is treated as a member of the first neuron's strong activation cluster, thus the $y$ attached to the first neuron will be used as the output. There might be no best universal choice of selective clustering, considering that the shape of local manifold may change depending on the dataset distribution. Hence, we leave it for further studies.

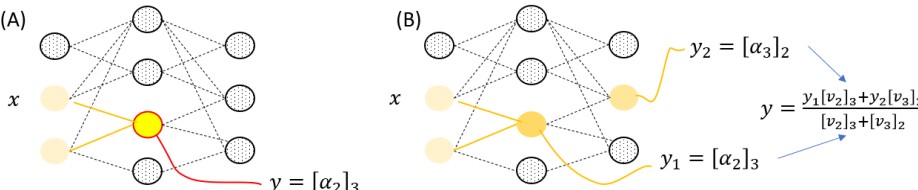

Figure 6: (A) Strong activation at layer 2 node 3 (yellow circle with red boundary). The output is taken as $[\alpha_2]_3$ (B) No strong activation, only 2 moderate activations (circles with darker shades of oranges). The output shown is a weighted average.

### A.5.2 COMPUTING OUTPUT VIA SQANN PROPAGATION (PREDICTION)

Fig. 6 shows two cases occurring during SQANN propagation Case (A) shows a case when strong activation occurs at layer 2 node 3, i.e. $[v_2]_3 > \tau_{act}$, near 1 or even exactly 1 (if $x$ is the exact training sample used during construction). Then $y = [\alpha_2]_3$. In case (B), no nodes are strongly activated. The $\alpha$ values of the two most strongly activated neurons are used for weighted average based on the strength of activations $[v_2]_3$ and $[v_3]_2$, i.e. interpolation is performed. Clearly, we can explore different variations, for example, taking three most strongly activated neurons etc.

Link back to main text section 3.

### A.5.3 GOOD PRACTICE FOR SCALABILITY

For both TNN and SQANN, if there are 1 million training data, there will be 1 million neurons. Do we need all 1 million data? Probably not, since it might be fair to assume that some data points in the training dataset may have similarities. This is where pre-processing can be done to ideally root out data that are redundant. SQANN itself can be used to check how redundant they are; for example, a subset of the data can be collected and a smaller SQANN can be constructed for testing purposes. We can check how the subset of 'similar data' activates each other's neuron within this mini SQANN.

Link back to *computing output via SQANN*: go to lemma 1.

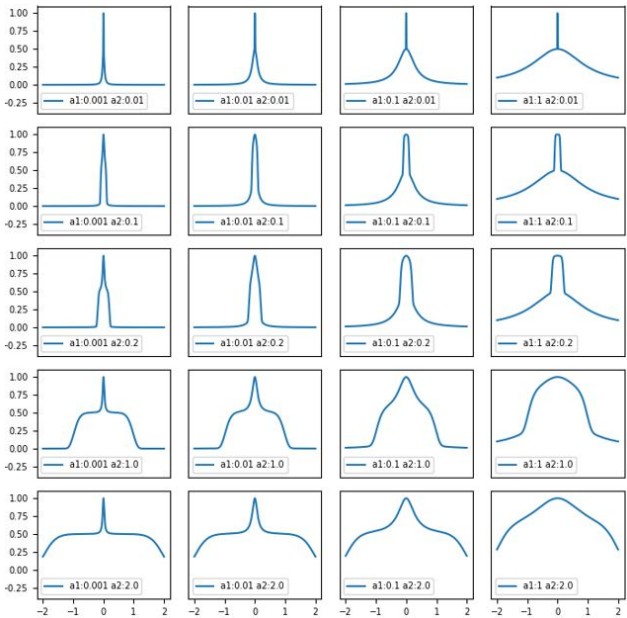

Figure 7: More $a_1, a_2$ variations of double selective activation.

### A.5.4 SQANN CONSTRUCTION CAN COMPLETE

**Proposition 3**. SQANN construction completes with high probability $p \approx 1$.

We start with the sketch of proof: the precise proof will need a strong assumption on data distribution. Note: we exclude any dataset which is ill-defined, i.e. when there exists two identical $x^{(k)} = x^{(k')}$ having different $y^{(k)} \neq y^{(k')}$. This results in unresolvable collisions, and is not suitable for any function.

Let the activation space $\mathcal{A}$ be a high dimensional space of pattern activations illustrated in fig. 8. An element of $a_{lk} \in \mathcal{A}$ is a collection of $[v_l]_k$, and distances between $a_{lk}, a_{l'k'}$ can be, for example, Euclidean. In SQANN construction, however, you might have noticed that the "distance" is expressed by eq. (5), i.e., only the between the activations at the relevant layer (during layer $k$ check for admission, it will be layer $k$; during collision, it will be the collision layer. Let red x be any activation $a_{lk}$ in $\mathcal{A}$ stored in SQANN currently being used to test a new sample in a *check $N_k$ activation* process during construction. Let black open circles be pattern activations of other sample points already incorporated into SQANN layers. Suppose we test a new training sample. From the main SQANN construction, we have seen that 3 possibilities can occur. (1) Admission of sample to layer $N_k$, if the sample weakly activates the existing node without collision, like the green + mark in fig. 8(B) (2) collision if it activates an existing node very strongly, like the red + mark (3) filtering into deeper layer if neither occurs, i.e. like blue + mark.

The concern related to the completion of SQANN construction mainly stems from the collision problem. During collision, layers are torn down, and the red + mark is now added beside the red x mark; thus SQANN now can distinguish similar looking points that might be characterized differently (*selective clustering* in action). The question is, will the black open circles be reconstructed back to the previously destroyed layers in the same manner as they were? The answer is, they will NOT be exactly the same as before since the new activations now need to consider a new node contributed by red + mark.

*Intuition behind $\sigma_{dsa}$, double selective activation.* However, there is a high probability that they will come back in the same order. This is the exact reason why double selective activation has a narrow band of strong activation (small red area in fig. 8) distinguished from a band of moderate activation by steep gradient, that is in turn distinguished from near zero activation by steep gradients. With such characteristic, $\mathcal{A}$ will have a small red region of strong activation, a large orange buffer region of moderate activation and weak activation everywhere else. The main idea revolves around the fact that the buffer region reduces the probability that existing black open circles are too close to red + mark. Hence, the activation characteristic of each black open circle that has been temporarily removed from the torn down layer (because of collision) will be similar to the activation characteristic it has before i.e. it will not suddenly strongly activate a node it previously did not activate. For example, if sample $u$ weakly activates all existing neurons with values 0.01 before, after red + mark is added into the layer, $u$ still weakly activates all existing neurons with values 0.01, including red + mark which is close to red x mark. More precisely, if $u$ gives rise to $v$ before collision (let us denote its component by $[v_i], i = 1, \ldots, n_l$, then after resolving collision, the new activation at the same layer is $v'$ such that $[v'_i] \approx [v_i], i = 1, \ldots, n_l$, although now $[v'_i]$ has the $(n_l + 1)$-th component since the collided sample has been added into the layer.

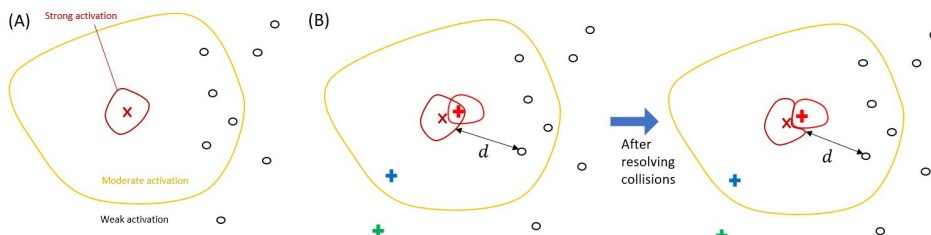

Figure 8: (A) Visualization of a SQANN node $\eta$ in the *activation space*, marked as red x. Black open circles are the fingerprints of other nodes that are already integrated into SQANN. (B) A data sample whose activation lies within the red/orange region such as red/blue plus mark is strongly/moderately activating the particular node $\eta$. Otherwise, it is weakly activating it, e.g. green plus mark. A dataset and SQANN construction settings are desirable if for every training data $u$, there is a reasonably large $d$ such that the probability that complication occurs is $p_u^c \approx 0$. When a data sample $x'$ strongly activates node $\eta$ (red plus mark), it will cause collision. If this happens, $x'$ will be integrated into the collision layer as $\eta'$, i.e. collision is resolved, hence shifting the *effective* shape of activation shape of $\eta$ shape (blue arrow). What used to strongly excite $\eta$ might now excite $\eta'$ more strongly (say, if it is nearer to $\eta'$).

*Probability of complication.* Moving on from the illustration, let any sample $u \in U$ where $U$ is the subset of dataset not yet used during SQANN construction. If its activation $v < \tau_{act}$ on all existing nodes (i.e. are not activating any existing nodes too strongly), then, if any new sample $s_1$ activates $s$ strongly, then $u$ does not activate $s_1$ strongly. Intuitively, this is because $s_1$ is near $s$, which is far from $u$ because there is a buffer region, thus *there is a high probability that $u$ is also far from $s_1$* (note: by *far*, we mean distance in the activation space). It is only with high probability, because *complication* may occur, as the following. In real dataset, there may be a non-zero probability that there are black circles lying at the edges very close to the boundary of red x mark and red + mark happens to activates that region. Let the probability that a complication occurs for $u \in U$ be $p_u^c$.

*Assumption for formal proof.* Formal result can thus be stated for $p \approx 1$ if we have an assumption on data distribution; recall from the statement of the proposition that $p$ is the probability that

construction is completed. Suppose $W, U$ are subsets of dataset that respectively have already and not already been used for SQANN construction . Let $A(x)$ be the set of points strongly activating $x$ (in the illustration, this is the area covered by the small red circle). Also let $d$ (illustrated in fig. 8(B)) denote the minimum distance between $u \in U$ and any points within $A(w)$ for any $w \in W$. The required assumption is: for a given $\tau_{act}, \tau_{ad}$ and the current ordering of dataset, there exists $d > 0$, such that for any $u \in U$, for any $w \in W$, if the minimum distance between $A(w)$ and $u$ is greater than $d$, then the probability of complication $u \in A(w_1)$ for any $w_1 \in W$ is $p_u^c \approx 0$.

*Assumption can be weak; defining the strong assumption.* With this assumption, each sample at risk of behaving differently after the addition of red + mark contributes to $1 - p_u^c$ probability of reconstructing the same SQANN layer previously torn down. But if there are many collisions and $p = \Pi_{u \in U}(1 - p_u^c) = 1 - \delta$, then $\delta$ might blow up easily, especially if there are many data points close to each other. In this case, what we really need to consider are (1) our selection of $\tau_{ad}, \tau_{act}$ might be unsuitable, since they cause too many overlaps. As a rule of thumb, decrease $\tau_{ad}$ and increase $\tau_{act}$ to enable constructions with less overlaps between samples' activations (2) there are too many similar data in the samples that could have been represented equally well with a fraction of available data. In this case, it might be better to remove some data from the samples and create smaller subsets for SQANN layer construction, which is beneficial, since the remaining data can be used for validations. With these two fixes, the collision distances between two data points are relatively increased, and, there might exist $d$ such that $p_u^c = 0$. The strongest assumption is thus when there exists $d$ such that $p_u^c = 0$ for all unused training sample $u$ at any time, i.e. assume no complication.

*Proof for $p = 1$ with the strongest assumption: assume there is no complication at each sample checking step.* Suppose layer $1, \ldots, k - 1$ have been constructed, and $j - 1$ nodes have been added to the latest layer $k$ that is being constructed. Recall and note the difference between $x^{(k)}$ and $x^{<k>}$. Suppose sample $x^{(j)}$ causes collision in layer $c \leq k - 1$. Then we tear down all layers after layer $c$; here is where we will use the *order integrity*: let $X_{temp} = \{x^{<r>}, x^{<r+1>} \ldots, x^{<j-1>}\}$ be the set of samples that have been returned to the list of unused indices in the same order they have been put into SQANN layer (likewise $Y_{temp}$) e.g. $(x^{<r>}, y^{<r>})$ is the first sample in layer $c + 1$. Then, *resolve the collision* by concatenating $(x^{(j)}, y^{(j)})$ to $(N_c, \alpha_c)$ of layer $c$. The reconstruction of subsequent layers will occur as the following. Check $x^{<r>}$ for admission, add it into layer $c + 1$, then checking $x^{<r+1>}$ the same way it was added through "layer $k$ construction" process in the main text, and so on up to $x^{<j-1>}$. They will be returned to their previous positions the same way they were added to the torn down layers. By the no-complication assumption, samples will not collide with $(x^{(j)}, y^{(j)})$. More verbosely, samples admitted via "admission to $N_k$" will again be admitted to the same $N_k$ at the same node, while sample admitted through collision with some node $\eta_{k'}^{<j'>}$ will collide with the same node, and *there is no collision with the node of activation corresponding to $(x^{(j)}, y^{(j)})$* because of no complication assumption. As a consequence, we can always proceed with $x^{(j+1)}$, even if there is a need for multiple reconstructions of layers. Thus, the construction can complete. $\square$

## A.5.5   SQANN Main Theorem

**Theorem 2**. Assume SQANN construction is completed. SQANN achieves arbitrarily high accuracy on a training dataset. Furthermore, it is resistant to catastrophic forgetting.

Proof: Let the training dataset be $D = X \times Y$. For any $x \in X$, perform SQANN propagation. If $x \in N_1$, then, by lemma 1, we obtain the arbitrarily high accuracy. If $x \in N_k, k > 1$, the heavy-lifting is in fact already done through the SQANN construction algorithm. By SQANN construction, there exist index $j$ and layer $k > 1$ such that either (1) the activation of $x$ undergoes *admission to $N_k$* as $\eta_k^{<j>}$ or (2) pushed into layer $N_k$ during collision. One of the two must occur, otherwise SQANN construction is not completed, contradicting the assumption. In either process, we have $v_{k-1} = \eta_k^{<j>}$ thus $[v_k]_j = \sigma_{dsa}(||v_{k-1} - \eta_k^{<j>}||) = \sigma_{dsa}(0) = 1 > \tau_{act}$ where recursive computation $[v_l]_i = \sigma_{dsa}(||v_{l-1} - \eta_l^{}||)$ for all $i = 1, \ldots, n_l$ is performed from layer $l = 1, \ldots, k$ with $v_0 \equiv x$. Since such strongly activated neuron exists at layer $k$, by SQANN propagation, retrieve $y = \alpha_k^{<j>}$ where $j = argmax_{j'}[v_k]_{j'}$. To ensure that $j$ is unique, we reasonably assume that there is no duplicate $x$ with different $y$ values that is admitted through collision, otherwise the dataset is ill-defined (as previously mentioned). The uniqueness is made possible because double selective activation is, by definition, has a peak with unique value 1 and in the locality of the peak, to each

side, it is one-to-one and onto, i.e. there is no interval of constant value around the peak. Finally, $j$ is indeed the index from argmax, because the peak value of double selective activation is 1 by design.

*Resistance to catastrophic-forgetting.* For each new training sample added into SQANN, previous samples still exist as nodes stored in the SQANN layers. Previously learned samples will therefore not be forgotten. More precisely, suppose SQANN construction is not yet complete. Let $D_k \subset D$ be the subset of dataset whose samples have either been admitted to layers $1, 2, \ldots, k$ through normal admission or by resolving collision. Then for $p_k = (x^{(k)}, y^{(k)}) \in D_k$ can be exactly queried by obtaining activation at some layer $l \leq k$ for some $j$ so that $v = \sigma_{dsa}(\|v_{l-1} - \eta_l^{<j>}\|)$ is exactly 1 and $l, j$ exist exactly at the layer and node where $p_k$ is admitted into SQANN through the main mechanism of layer construction. By SQANN propagation, as before, we retrieve the output $\alpha_l^{<j>} = y^k$ stored exactly at $l, j$ node through the main SQANN construction algorithm as well, hence, SQANN remembers the previously stored value. Furthermore, if SQANN construction with $D$ has been completed and new training dataset $D'$ is available, construction with all samples in $D$ by drawing samples in the same order will result in the same SQANN. Any new samples from $D'$ can be admitted into SQANN in the same way specified in *layer $k$ reconstruction after* all $D$ samples are used up. In this manner, previously learned samples from $D$ will remain inside SQANN layers, i.e. like before we can find $l, j$ node corresponding to each sample previously admitted into SQANN. Hence no forgetting will occur. $\square$

### A.5.6 SQANN EXAMPLE

**SQANN pencil-and-paper example**. With $a_1, a_2 = 0.001, 0.5$, $\tau_{ad}, \tau_{act} = 0.1, 0.9$, create SQANN universal approximator for indexed data $X = [x^{(1)}, x^{(2)}, x^{(3)}, x^{(4)}] = \begin{bmatrix} 1 & 1.2 & -1 & -1.2 \\ 1.2 & 0.8 & -1 & -1.2 \end{bmatrix}$ and $Y = [y_1, y_2, y_3, y_4] = [1, 1, 0, 0]$. (A) Show layer 1 stores the fingerprints of $(x^{(1)}, x^{(3)})$ and layer 2 stores $(v^{(2)}, v^{(4)})$ i.e. activations of $(x^{(2)}, x^{(4)})$. (B) Use SQANN propagation to verify that we indeed get zero errors on $X \times Y$. (C) Test SQANN on the test dataset $X_{test} = [x_t^{(1)}, x_t^{(2)}, x_t^{(3)}] \begin{bmatrix} 1.25 & -1.25 & -1 \\ 1.25 & -1 & -1.4 \end{bmatrix}$ and plot the results, marking the interpolations made by SQANN.

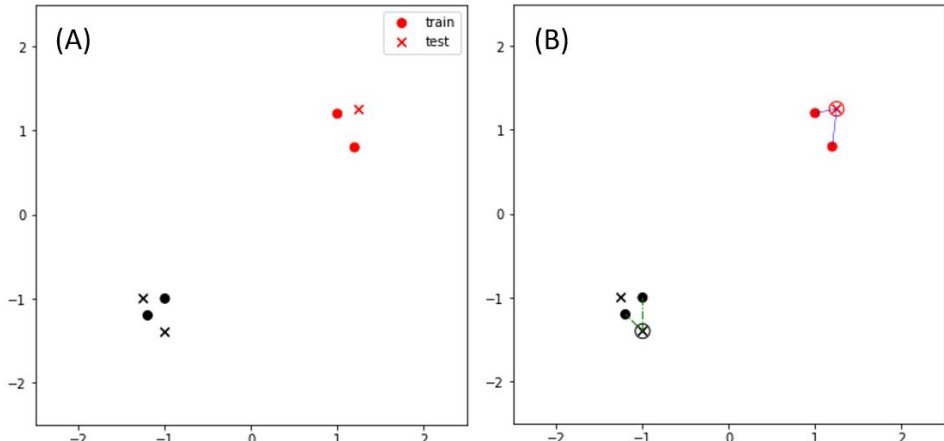

Figure 9: (A) Training samples are in closed circles, test samples are in x. (B) Open circles show samples that need interpolation, and the corresponding lines point towards the samples from which they are interpolated.

Note that the following can be matched with the demonstration in jupyter notebook SQANN_small_example.ipynb.
(A) *SQANN construction.* We start by putting $x_1, y_1$ into the first layer, so put it into the first layer of SQANN, $N_1 = (x_1), \alpha_1 = (y_1)$. Now check $(x^{(2)}, y^{(2)})$ for admission to $N_1$: if it activates a node in $N_1$, then we *filter it to a deeper layer*, otherwise, we add it into $N_1, \alpha_1$ as well. We show that the latter occurs. We only have one node in SQANN now, so the only possible activation is $[v_1^{(2)}]_1 = \sigma_{dsa}(\|x^{(2)} - x^{(1)}\|) = 0.3344 \geq \tau_{ad}$. Admission to $N_1$ only occurs if $[v_1^{(2)}]_1 < \tau_{ad}$ hence it is filtered to a deeper layer.

Now we check $(x^{(3)}, y^{(3)})$ for admission to $N_1$ and get $[v_1^{(3)}]_1 = 5.655 \times 10^{-5}$. It does not activate any node in the layer, thus this is a distinct sample we will admit into $N_1$.

Now we check $(x^{(4)}, y^{(4)})$ for admission to $N_1$. We have two nodes, so we have to compute both: $[v_1^{(4)}]_1 = \sigma_{dsa}(||x^{(4)} - \eta_1^{<1>}||) = 4.450 \times 10^{-6}$ and $[v_1^{(4)}]_2 = \sigma_{dsa}(||x^{(4)} - \eta_1^{<2>}||) = 0.5676 > \tau_{ad}$. It does activate a node in $N_1$, which is $\eta_1^{<2>}$. Hence it is filtered to a deeper layer. We have shown that $N_1$ stores $(\eta_1^{<1>} = x^{(1)}, \eta_1^{<2>} = x^{(3)})$, and not the other samples. Note: since the activation $< \tau_{act}$, no node has been activated *strongly* (otherwise we will have collision). Recall that each time we store $x^{(k)}$ into $N_l$, we also store $y^{(k)}$ into $\alpha_l$, so now $\alpha_1 = (y^{(1)}, y^{(3)})$.

We have gone through the training dataset once. The unused data are $\{(x^{(k)}, y^{(k)}), k = 2, 4\}$, as they have been filtered to deeper layer. We now proceed with layer 2 construction. Since it is empty, we put the activation of $x^{(2)}$ (not the sample itself) into $N_1$ since no collision occurs, i.e. $\eta_2^{<1>} = v_2^{(2)} = [[v_2^{(2)}]_1, [v_2^{(2)}]_2]$. To show no collision, i.e. no strong activations in $N_1$, similar to before, we compute $[v_1^{(2)}]_1 < \tau_{act}$, which is previously done, and $[v_1^{(2)}]_2 = 6.187 \times 10^{-5} < \tau_{act}$. The last sample $x^{(4)}$ is admitted into $N_2$ by checking collisions against all $N_1$ (previously done), and then checking activation against existing $N_2$ node, $\eta_2^{<1>}$, i.e $v_2^{(4)} = \sigma_{dsa}(||v_2^{(4)} - \eta_2^{<1>}||) = 0.00732 < \tau_{ad}$. Hence, it is admitted to $N_2$. We have used up all data points, hence we have shown $N_2 = (v_2^{(2)}, v_2^{(4)})$.

(B) Using SQANN propagation on $x^{(k)}, k = 1, 3$, we get $\sigma_{dsa}(||v_1^{(k)} - \eta_1^{<j>}||) = \sigma_{dsa}(0) = 1 > \tau_{act}$ where $j = 1, 2$ respectively. Since they are strongly activated, we get $y = \alpha_1^{<j>} = y^{(1)}, y^{(3)}$ for $j = 1, 2$ respectively. Thus the errors are zero.

For $x^{(2)}$, we already previously computed $v_2^{(2)} = [[v_2^{(2)}]_1, [v_2^{(2)}]_2]$, which becomes $\eta_2^{<1>}$ thus we will also get the distance of activation value from itself as stored in $N_2$, $\sigma_{dsa}(||v_2^{(2)} - \eta_2^{<1>}||) = \sigma_{dsa}(0) = 1$. Likewise $x^{(4)}$.

(C) *SQANN testing*. For $x_t^{(1)}$, we expect its fingerprint to be close to $x^{(1)}$ or the activation of $x^{(2)}$ since their values are similar. It turns out we get the following activations in layer 1, $v_{t,1}^{(1)} = [0.5053, 4.938 \times 10^{-5}]$, not strongly activating layer 1 nodes. For layer 2, we have $[v_{t,2}^{(1)}]_1 = \sigma_{dsa}(||v_{t,1} - v_2^{(2)}||) = 0.5165$ and $[v_{t,2}^{(1)}]_2 = 0.0009896$. There is no strong activation anywhere, and this is the case when interpolation is needed. Notice that of all the activations we computed, the strongest are $[v_{t,1}^{(1)}]_1$ and $[v_{t,2}^{(1)}]_1$, which are the first nodes in both layer, due to $x^{(1)}$ and $x^{(2)}$ respectively as we have expected. Using SQANN propagation, we fetch its $\alpha$ values, both of which are 1., thus using linear interpolation, $y = \frac{0.5053*1 + 0.5165*1}{0.5053 + 0.5165} = 1$.

The next sample in the test dataset $x_t^{(2)}$ does activate the SQANN. We expect it to activate either $x^{(3)}$ or the activation of $x^{(4)}$. First, get $v_{t,1}^{(2)} = [5.049 \times 10^{-5}, 0.5059]$, i.e. no strong activation, so $x^{(3)}$ is not strongly activated. But we get $\sigma_{dsa}(||v_{t,1}^{(2)} - \eta_2^{<2>}||) = 0.9880 > \tau_{act}$, thus we do get strong activation due to the fingerprint of $x^{(4)}$. By SQANN propagation, we get $y = \alpha_2^{<2>} = y^{(4)} = 0$.

The last test sample needs interpolation too, and we leave it for the reader. The plot of results are shown in fig. 9.

### A.5.7   EXPERIMENTAL DATA

For all experimental data, we perform no pre-processing beyond simple normalization. We test all the data in nearly their raw forms.

**Experimental data** in fig. 4(A). [Top row] Training Training samples $X$ are $x \in \mathbb{R}^2$ with components drawn from uniform r.v. (random variable) $x_1 = x_2 = t \sim U(-1., 1.)$ plus Gaussian noise. Test samples are $X$ plus r.v. from $U(-s, s)$ where $s$ is the test data spread. Also, $y = ||x||$ is indicated by the color. [Bottom row] similar to top row, but $x \in X$ with $x = R[cos(t), sin(t)]^T$ where $t \sim U(0, \approx 2\pi), R \sim U(0.8, 1.2)$ and $y = cos(t)$. For both experiments in the top and bottom rows , 128 training data samples are used for SQANN construction and tested on 128 test data samples.

**SQANN is tested for regression on Boston Housing and Diabetes Datasets** to demonstrate its generalizability to unseen (test) samples. The procedure is as the following. Let Boston Dataset be $\mathcal{D} = \{(x^{(k)}, y^{(k)}), k = 1, \ldots, 506\}$. (1) A small set of samples $D = \{(x^{(k)}, y^{(k)}), k = 1, ..., 100\}$ is used to train SQANN and 9 other regression methods; recall that ordered sequence of data

Table 2: Comparing MSE on different regression methods for Diabetes dataset. Notations similar to table 1.

|  | Lin | Ridge | Lasso | LSVR | NuSVR | SVR | DTree | kneigh | MLP | SQANN |
|---|---|---|---|---|---|---|---|---|---|---|
| o. | 58.57 | 57.60 | 59.21 | 94.65 | 84.34 | 81.04 | 76.73 | 66.95 | 108.86 | 93.80 |
| e40 | 54.15 | 54.18 | 55.65 | 70.92 | 74.39 | 73.57 | 40.52 | 53.06 | 54.17 | 53.96 |

is used for SQANN construction. (2) Mean Squared Errors (MSE) values are measured on unused data $D_{test} = \mathcal{D} \backslash D$ on all 10 models listed in table 1. We expect large errors on some test samples, since the training dataset might be too small to be representative. (3) SQANN's activations are used to collect samples with large absolute errors $e_\tau(x) = |SQANN(x) - y_0| > \tau$ and we treat them as out-of-distribution (OOD) samples. These samples are considered as new distinct samples to be integrated into $D$ as the new training dataset $D'$. (4) Train the 10 models, now with $D'$. (5) Then MSE is measured again on $D_{test}$ (yes, there will be partial overfitting). Repeat the process for diabetes dataset, where $|\mathcal{D}| = 442$; we also start with $|D| = 100$.

Results for Boston Housing dataset has been described in the main text.

For Diabetes Dataset, $\tau = 40$ yields relatively competitive performance. using SQANN we integrated 218 samples, so $|D'| = 318$ so 0.719 of the whole $\mathbb{D}$ is used for new training. From the results of linear models, the dataset appears to have a somewhat strong linear structure; but non-linear models still can perform better.

## B  MORE COMMENTS

This section consists of contents that we have considered less important, thanks to comments by reviewers in ICLR 2022.

### B.1  ON DIRECT GENERALIZATION OF TNN

As mentioned in appendix section A.4.5, *our attempts at generalizing TNN directly to high-dimensional input data are unsatisfactory. Readers can skip this without losing any information required for understanding this paper.* For the record, we still list these attempts here. Now, we consider multi-dimensional input $x \in \mathbb{R}^n, n > 1$ with one dimensional output $y$ for TNN. If the well-known *space-filling curve* can be constructed for the input data, generalization is immediately done. Otherwise, variable separability might help with modelling a system partially using TNN. For example, given an ideal two-variable model $F(x, y) = f(x)g(y)$. Suppose it fits the experimental data poorly. Assuming that $f$ is correct, correction can be applied to $g$ by replacing it with $g(y) + NN(y)$ to account for the errors. Computing $TNN(y) = \frac{F(x,y)}{f(x)} - g(y)$ yields the $y$ values to be used for weight computations in our triangular construction. Clearly, error correction can be performed on $f$ similarly, though we now have to determine how to allocate values to each factor. One way to order multi-dimensional data samples will be ordering by the magnitude of the vector $x = [x_1, ..., x_N]^T \rightarrow [r, x_2, ..., x_N]^T$ where $r = \sqrt{\Sigma_i x_i^2}$. The sign of $x_1$ can be generally dealt with once we include discrete variables (such as binary variables) in the vector, where weights could be toggled to different values according to the discrete variables. More generally, weights that vary with a continuous variable may be a powerful modification to the current method of construction which uses constant weights. This will lead to the loss of linear ordering, in the sense of "larger" vs "smaller" in the inequality of real number, leaving us with more relaxed conditions, possibly partial order. However, they will not be within the scope of this paper. For now, assume we have already defined a linearly ordering for the dataset, we can now generalize the construction. Now $W \in \mathbb{R}^{N \times n}$, where $N, n$ are still respectively the number of data samples and dimensions. Similar to the earlier sub-section *computing weights*, $(Wx^{(k+1)} + b)_{N-k+1} = -a$ and $(Wx^{(k)} + b)_{N-k+1} = a$. By subtracting them, we now get $[W(x^{(k)} - x^{(k+1)})]_{N-k+1} = 2a$. Spelling it out and rewriting component-wise difference as $\Delta_i^{(k)} = x^{(k)} - x^{(k+1)}$, we have

$$\Sigma_{i=1}^n W_{N-k+1,i} \Delta_i^{(k)} = 2a \quad (6)$$

Since $\Delta_i^{(k)}$ can be zero, we only need to carefully choose $W_{N-k+1,i}$ to satisfy equation (6). This is simply done by setting

$$W_{N-k+1,i} = \begin{cases} 0 & if \ \Delta_i^{(k)} = 0 \\ \frac{2a}{N_1 \Delta_i^{(k)}} & otherwise \end{cases} \tag{7}$$

where $N_1$ is the number of non-zero $\Delta_i^{(k)}$ over all $i$ for the particular $k$. Finally, $b_{N-k+1} = a - W_{N-k+1}x^{(k)}$ and likewise $\alpha = A^{-1}y$ in the same manner.

The difficulty is in selecting the choice of the linear ordering. We are not able to provide any sensible ordering for high dimensional dataset, though a meaningful ordering we may still exist. Thus we ask does there exist any linear ordering so that TNN can be used for high accuracy classification? The proposition indicates that there is.

**Proposition 4** . *Given a standard DNN for C classes classification with $a_{tr}$ training accuracy, then there exists a linear ordering for TNN to achieve $a_{tr}$ accuracy. Test accuracy $a_{test}$ of DNN can be achieved by TNN with high probability through squeezed linear ordering.*

Proof: In the stringent case, classification is performed by taking $c = argmax_i y_i$ where $y = DNN(x)$, $x$ the image to be classified, and $y \in \mathbb{R}^C$ the output from the last layer of the neural network $DNN$. Suppose $C = 10$ and $c = 0, 1, \ldots, 9$. Partition the unit interval $[0,1]$ such that the ten classes are evenly distributed, i.e. $I_c = [\frac{c}{C}, \frac{c+1}{C}]$. For each class $c$, collect all the training data samples that are classified as $c$ by the DNN, denote this set as $P_c$. Then perform the following linear ordering: map $p_c$ to $\frac{c+0.5}{C}$ where $p_c$ is the data point with the highest $y_c$ component at the last layer of DNN. Let $R_c = \{r_i \in P_c \backslash \{p_c\} : DNN_{c+1}(r_i) \geq DNN_{c-1}(r_i)\}$. Then, continue constructing the linear ordering $p_c < r_1 < r_2 < \ldots$ so that $r_k < r_{k+1}$ whenever $DNN_c(r_k) \geq DNN_c(r_{k+1})$, i.e. going to the right of the interval, a data sample has the less probability of being classified as $c$ and it is also more likely classified as $c+1$ than $c-1$ if $c$ is excluded. Let $L_c$ be the remaining set of data samples classified as $c$, i.e. $P_c \backslash (R_c \cup \{p_c\})$. Set $\cdots < l_2 < l_1 < p_c$ so that $l_{k+1} < l_k$ whenever $DNN_c(l_k) \geq DNN_c(l_{k+1})$. The linear ordering for training dataset is done and TNN will yield $a_{tr}$.

*Squeezed linear ordering.* We can map the ordered samples to the interval $I_c$ arbitrarily, so long as the order is preserved. However, placing each $R_c, L_c$ nearer to $p_c$ increases TNN chances of achieving $a_{test}$, i.e. we squeeze the linear ordering towards the centers $p_c$ for all $c$. This is because, ordering test data samples into the linear order made by training samples, we may get the situation where test sample has max $y_c$ (thus likely classified as $c$), but lies beyond the extreme end of $R_c$ or $L_c$. Conversely, such sample is beyond the extreme end of $L_{c+1}$ or $R_{c-1}$ respectively. By squeezing, large spaces (the *grey area*) are reserved between the extreme ends of two classes. Data samples could fall into these intervals, and we reduce the possibility of misclassifying edge cases, e.g. by letting users treat them separately when TNN indicates that these samples lie in the *grey area*. □

Since TNN can achieve arbitrary accuracy on the training dataset, we will recover the same accuracy as the DNN from the newly ordered training data. We do not quantify the exact probability of attaining high $a_{test}$ because distribution of test dataset may not be exactly the same as the training dataset. With DNN as the encoder mapping the raw data into the unit interval is ironic since the hardwork is already spent on training the deep neural network. However, this shows that, in principle, there exist encodings that can achieve linear ordering for TNN to attain high accuracy classifications. Notes: when clashes occur, e.g. $r_k$ and $r_{k+1}$ have the same $DNN_c$ components, for the purpose of this simple construction, simply randomly assign order between them. We only aim to prove the existence of linear ordering here.

## B.2 More remarks on SQANN

**Regarding the pseudo code**. Mapping names from pseudo code to python code:
sample_collection $\rightarrow$ layer_k_sample_collection.
push_node $\rightarrow$ push_node_to_layer.
forward_cons $\rightarrow$ forward_to_layer_k_for_reconstruction.
ssig $\rightarrow$ STOP SIGNAL

collision → COLLISION.

remove_index → remove_index_to_layer_node check_signal, new_nodes and update_nodes are placeholders for simple check and update sequences in python code.

