# OpenReview forum: "Two Instances of Interpretable Neural Network for Universal Approximations"
_ICLR.cc/2022/Conference — ICLR 2022 Submitted_

### Official Review · Reviewer_v56a · 2021-10-22

**Correctness:** 2
**Technical Novelty And Significance:** 2
**Empirical Novelty And Significance:** 2
**Recommendation:** 3
**Confidence:** 4

**Main Review:**

Strengths:
1. This paper provides two new proof of the universal approximation theorem by construction. To my knowledge, the constructions are novel and interesting.

Weakness:
1. The writing should be improved for proofreading.
1.1. Some figures are far away from the paragraphs describing them. For example, the double selective activation is defined on page 5, while the illustration of this activation appears on page 3 (Fig 1(A)).
1.2. Mathematic formula should be more professional. In the 7th line of section 2, x^(k) are n-dimensional vectors and the relation "<" is not defined for them. Texts in math formula should not be oblique, such as "otherwise", "argmax", and "if".
2. The claimed contributions are minor or not supported.
2.1. Interpretable constructions for the universal approximation is not novel. Chapter 4 of [1] provides a visual (and can be proven rigorously) proof of the universal approximation, which indicates that neural networks can approximate functions using "almost" piecewise constant functions.
2.2. The proof of the universal approximation using TNN only holds for 1-dimensional inputs, as admitted by authors in the paragraph begins with "generalization to n-dimensions" at the end of page 4.
2.3. The claimed “universal approximation” is not exactly the one machine learning cares. This paper only considers fitting training samples, while the universal approximation in machine learning considers approximating a target function on a compact set.
2.4. The resistance to catastrophic forgetting is not supported by theorems. Authors claim that this contribution is supported by Proposition 1 on page 4, but Proposition 1 has nothing to do with catastrophic forgetting. Catastrophic forgetting is " the tendency for knowledge of previously learned task(s) (e.g. task A) to be abruptly lost as information relevant to the current task (e.g. task B) is incorporated"[2], which needs two sequential tasks. Although D and DUA are two training sets, Proposition 1 has nothing to do with catastrophic forgetting in the following sense: i) DUA contains D as a subset, thus the knowledge in the previous task (D) also exists in the current task and forgetting does not exist obviously; ii) the statement of Proposition 1 is trivial since we can always choose A=D’ and the conclusion in Proposition 1 is the same as Theorem 1.
2.5. The generalization is not the one machine learning cares about. The paragraph begins with “generalizability to test dataset” on page 4 claims that “they (test data) can be incorporated into the training dataset to create a better model with the above-said error upper-bound”. In machine learning, the label of test data should not be used for training.

[1] Nielsen, M. A. (2015). Neural networks and deep learning (Vol. 25). San Francisco, CA: Determination press.
[2] Kirkpatrick, J., Pascanu, R., Rabinowitz, N., Veness, J., Desjardins, G., Rusu, A. A., ... & Hadsell, R. (2017). Overcoming catastrophic forgetting in neural networks. Proceedings of the national academy of sciences, 114(13), 3521-3526.


**Summary Of The Paper:**

Summary:
1. This paper proposes two neural networks by construction, i.e., Triangularly-constructed Neural Network (TNN) and Semi-Quantized Activation Neural Network (SQANN).
2. These two neural networks are universal approximators, which is proven by construction.
3. These two neural networks are resistant to catastrophic forgetting.
4. For TNN, strongly activated neurons and half-activated neurons can be identified.
5. For SQANN, users can identify the samples that are likely out of distribution.

Contributions claimed by authors:
1. TNN and SQANN are proposed.
2. The universal approximation is proven, using the construction of TNN and SQANN.
3. Resistance to catastrophic forgetting is proven.
4. SQANN can identify out-of-distribution samples.


**Summary Of The Review:**

The contributions are not novel or even not supported.
1. Interpretable universal approximation is not novel. It is written in a book in 2015.
2. The resistance to catastrophic forgetting is not supported by theorems. Settings of the claimed theorems are different from that of catastrophic forgetting.
3. The universal approximation and generalization are not the ones machine learning cares about.

---

> ### Author Response · Authors · 2021-11-09
> **Author's Response**
>
> Thanks for taking the time to review the paper. We will revise our paper. The following is our attempts to address the feedback you have given.
>
> We will for now ignore comments on proofreading: thank you for the advice, we will readjust them.
>
> **1. Comment**: Chapter 4 of [1] provides a visual (and can be proven rigorously).
>
> *Response*: Yes, but is there any rigorous proof in Chapter 4 of [1] as of 9 November 2021? Unlike that website, we do prove it rigorously. We provide explicit construction, not just sliding bar demonstration.
>
> **2. Comment**: The proof of the universal approximation using TNN only holds for 1-dimensional input
>
> *Response*: yes, which is why we ‘upgrade’ it to the SQANN.
>
> **3. Comment**: The claimed “universal approximation” is not exactly the one machine learning cares.
>
> *Response*: this is very badly said. Does reviewer’s comment “while the universal approximation in machine learning considers approximating a target function on a compact set” imply that approximating functions in other ways are not allowed? To be honest, such response is highly disappointing.
>
> **4. Comment**: Authors claim that this contribution is supported by Proposition 1 on page 4, but Proposition 1 has nothing to do with catastrophic forgetting.
>
> *Response*: reviewer quotes “the tendency for knowledge of previously learned task(s) (e.g. task A) to be abruptly lost as information relevant to the current task (e.g. task B) is incorporated"[2]” in a way that seems to imply an absolute definition. We find its usage to deny our proposition highly questionable. If we assert that our version of catastrophic forgetting is “the tendency for knowledge of previously learned dataset to be abruptly lost as information relevant to the new dataset is incorporated”, are we wrong? We believe this is a viable definition too.  Furthermore, proposition 1 is NOT trivial, because incorporating “out-of-distribution” samples into the model is a real problem.

---

> > ### Comment · Reviewer_v56a · 2021-11-10
> > **Some Arguments**
> >
> > 1. The rigorous proof of Chapter 4 in [1] is not difficult and can be completed by most undergraduate students. The rigorous proof is novel but the contribution is not enough to publish a paper on a top conference.
> >
> > 2. This point should be clearly clarified at the beginning of chapter 2, where $x \in [0,1]^n$ is confusing.
> >
> > 3. Comments are divided into two parts, where 3.1 is about academic standards and 3.2 is about the "universal approximation theorem" in this paper.
> >
> > [3.1] The term "universal approximation" or "universal approximation theorem" is a widely used terminology since 1990s. If authors must use exactly the same terminology, it is necessary to: i) clearly clarify the difference between its original meaning and the new meaning in this paper; ii) explain why the new meaning is worth studying and what is the advantage of this new meaning.
> >
> > [3.2] The "universal approximation" in this paper is of few theoretical significance. The original universal approximation concerns about the approximation capabilities of neural networks and universal approximation theorems guarantee the existence of a neural network which expresses the target function with small error. However, the "universal approximation" in this paper cares about the accuracy on the training dataset. It is widely known that low training error is not the goal of training a model since the model may overfit the training set. The theoretical value of the "universal approximation" here is not clearly demonstrated.
> >
> > 4. Comments are divided into two parts, where 4.1 is about academic standards, 4.2 is about authors' version of "catastrophic forgetting", and 4.3 is about the triviality of Proposition 1.
> >
> > [4.1] Similar as 3.1.
> >
> > [4.2] What is the "knowledge of previously learned dataset" in Proposition 1? Both the definition of authors' "catastrophic forgetting" and its relationship with Proposition 1 are poorly illustrated in the paper.
> >
> > [4.3] Proposition 1 is trivial, since A can be the same as D'. When A is chosen as D', TNN is constructed on D and D', i.e., the training set is D and D'. Then Theorem 1 indicates that TNN achieves arbitrarily high accuracy on D and D', which is a stronger conclusion than the that of Proposition 1. Proposition becomes meaningful only if more restriction about A is added. Authors argue that incorporating ood samples into the model is a real problem. The following should be noticed: i) what is ood samples in Proposition 1? Maybe D' is ood samples, but D' is part of training set in the current task; ii) triviality here means that the proof of Proposition 1 is obvious from Theorem 1 since A can be chosen as D', and triviality is not about real problems.

---

> > > ### Author Response · Authors · 2021-11-10
> > > **Revision based on your update**
> > >
> > > Thank you for sparing more time to reply to our comments. Very generous of you, actually.
> > >
> > > We will include explicit **Concept disambiguation** section at the intro of our paper now to answer comments [3.1], [3.2], [4.2].
> > >
> > > Okay, I think I see something interesting here. There are concepts that are of very little concerns to theorists, thus they are not of clear utility. For example, in [4.3], it is clear that including all data $D\cup D'$ ie. $A=D'$ will give us perfect accuracy. The computational cost will be the issue here. But suppose there are only 1% ood samples D', we will have a more meaningful set A. Including all D' will be wasteful, in a sense. Maybe this is what you mean by 'only if more restriction about A is added'. In real life, can we constrain test dataset behaviour in the wild? If we can, it is a good experimental bonus, I guess, or the system is very well behaved.
> > >
> > > Anyway, thanks for the comments on TNN. We probably have a whole host of problems in SQANN for you to roast too, feel free to do it.

---

### Official Review · Reviewer_KGST · 2021-10-29

**Correctness:** 2
**Technical Novelty And Significance:** 1
**Empirical Novelty And Significance:** 1
**Recommendation:** 1
**Confidence:** 5

**Main Review:**

A very similar method for "constructing" neural network was presented by Szymanski.etal "Deep, super-narrow neural network is a universal classifier".  As far as I can tell, the requirement of enforcing an ordering of training point is equivalent to  Szymanski's projection of the data onto a line...which in effect order the points...and the manner of dividing the points into classes by stacking classifies side by side along the projection seems similar.  That paper shows that the choice of ordering of the inputs has a massive effect on generalisation.  So my question is - how is the choice made in the method proposed in this paper?  Authors claim that ordering of inputs gives rise to interpretability (though it is not clear to me how exactly that works)...but if the ordering is arbitrarily chosen by the user, then this gives nothing more than what the user already knows about the data (when making that ordering choice).

As far as I can tell, the catastrophic forgetting resistance claim boils down to ability of the model to perform correct approximation on a training dataset supplemented with new data points.  Is this understanding corrct?  Doesn't that imply that some points (presumably at least the two before and after the new point in the sequence) need to be known for the adjustment?  Does this involve adding more architectural elements (presumably at least one more neuron) to the architecture?  Catastrophic forgetting is about ability of a model to learn a new task, often out of distribution of the first task, without having access to training data of the previous task.  Not sure how the arguments presented in this paper relate to catastrophic forgetting.

I am very confused by the claim of generlisation of the proposed model.  My reading of it is that authors show that the error on a test point is bounded, but not by an arbitrarily small number, and the suggestion is to incorporate the test points that have high error into the train data.  Is this high level assessment of the presented proof correct?  If that's so, then wouldn't the model "generalise" best if all test points were incorporated into the training set....and wouldn't every model out there "generalise" by this standard by simply including the points with test error into the training data?



**Summary Of The Paper:**

The paper presents a method for constructing neural networks where the architecture and the weights are derived analytically from the structure of ordered inputs instead of iterative training to provide arbitrary accuracy on training data.  Authors make a number of unconventional arguments about their architecture being resistant to catastrophic forgetting, and generalisation ability of the proposed model.

**Summary Of The Review:**

The proposed model of progressive constructive of neural networks does not seem to be novel and has been shown to provide universal approximation as well as have limits, which the authors of this method do not seem to be aware of.  While the resulting architecture is capable of arbitrary accuracy of approximation on the training data, I don't see that it would generalise or cope with catastrophic forgetting, at least according to the typical notions of what constitutes generalisation and catastrophic forgetting in machine learning.

---

> ### Author Response · Authors · 2021-11-09
> **Author's Response**
>
> Thanks for taking the time to review the paper. We will revise our paper. The following is our attempts to address the feedback you have given.
>
> **1. Comment**: “A very similar method for "constructing" neural network was presented by Szymanski.et al …   Szymanski's projection of the data onto a line...which in effect order the points... and the manner of dividing the points into classes by stacking classifies side by side along the projection seems similar.
>
> *Response*: ‘Very similar’ is an overstatement.   TNN might be a very special case of Szymanski's, but SQANN is an upgrade of TNN along a fundamentally different direction. For example, in Szymansk’s, “the representational power of the super-narrow network is solely derived from depth and not breadth”. This is not true for SQANN. In fact, for SQANN to leverage depth rather than breadth, we have to adjust the admission threshold to be so low that many points will end up in the vicinity of earlier sample points to be pushed into deeper layer.
>
> **2. Comment**: “That paper shows that the choice of ordering of the inputs has a massive effect on generalisation… - how is the choice made in the method proposed in this paper?
>
> *Response*: for TNN, we assume the ordering is already there defined by human, thus it is not useful, as the reviewer said. Yes, this is right. But *SQANN does not require a pre-defined ordering*. In fact, the order of the data is useful as an indexing ‘tool’. If a sample activates a neuron that corresponds to data $x^{(10)}$, then we know that sample is similar to $x^{(10)}$. However, the order of the data can be anything; SQANN’s job is precisely to check one data after another and arrange them into the layer meaningfully. In a sense, SQANN is providing an ordering in the form of stacked layers.
>
> **3. Comment**: Authors claim that ordering of inputs gives rise to interpretability (though it is not clear to me how exactly that works)....
>
> *Response*: for TNN, ordering gives interpretability: see the example we mentioned in the conclusion about ECG. A time series of size 100 can be pre-processed, for example be extracting the 10 useful points that correspond to meaningful landmark in time, such as the P, Q, R, S or T points in the ECG signals.
>
> **4. Comment**: (1) “the catastrophic forgetting resistance claim boils down to ... Is this understanding correct?” (2) Doesn't that imply that some points … need to be known for the adjustment? (3) Does this involve adding more architectural elements (presumably at least one more neuron) to the architecture?
>
> *Response*: (1) Yes, correct in our context. (2) the question is only applicable for TNN. Yes, we need to somehow place the new point between the two points in the sequence. (3) yes, for both TNN and SQANN. In SQANN, we do not need to know the previous training dataset, since it is actually processed into a “neuron” within the model. Yes, ‘adding more architectural elements’ is also a property of our interpretable neural network, and we have made an analogy with mammalian brain which ‘adds more elements’ by growing dentrites.
>
> **5. Comment**: “I am very confused by the claim of generlisation of the proposed model… the suggestion is to incorporate the test points that have high error into the train data. (1) Is this high level assessment of the presented proof correct?” (2) then wouldn't the model "generalise" best if all test points were incorporated into the training set
>
> *Response*: (1) yes, the high level assessment is correct. (2) YES, exactly. In fact, in the extreme case where test points are all out of distribution, it is only natural to include them, although not including them does not imply that our TNN immediately fails -we can choose not to include them at the expense of a little error, or no error at all. Otherwise, in reality, some data points resemble data points we use for training. Some data within the training datasets themselves should be excluded (if possible, and used as validation points instead), so that we do not have unnecessary copies of neurons in SQANN models. Even then, having two copies of very similar points included are not necessarily bad, though multiple copies of them might be costly. Hence, for our model, finding the ‘representative samples’ might be the correct way of pre-processing data.
>
> This reviewer also quotes the definition of Catastrophic forgetting; see our response no. 3 to the comment of reviewer v56a.

---

> > ### Comment · Reviewer_KGST · 2021-11-10
> > **Re: Author's Response**
> >
> > Thank you for providing responses to my questions.  However, having my understanding of the presented concepts confirmed, it becomes clear that the capabilities of the proposed models boil down to ability of a perfect fit on the training data, and not much else.  Unfortunately, for the purposes of generalisation, catastrophic forgetting and interpretability, as they are defined in the machine learning community, this characteristic alone, is of little use and interest.

---

> > > ### Author Response · Authors · 2021-11-11
> > > **Is the Reviewer denying the whole premise of Universal Approximation?**
> > >
> > > Is the Reviewer denying the whole premise of Universal Approximation?
> > >
> > > Maybe yes, maybe no. It's very hard to tell.
> > >
> > > Oh, right, the score of the paper is so low it's going to get rejected anyway. Regardless, for completeness sake, we're going to submit a revision.
> > >
> > > Thanks for sparing the time to review the paper, really appreciate it.

---

### Official Review · Reviewer_Xw97 · 2021-10-29

**Correctness:** 3
**Technical Novelty And Significance:** 1
**Empirical Novelty And Significance:** Not applicable
**Recommendation:** 3
**Confidence:** 3

**Main Review:**

Strengths:

The idea of making neurons reflect training data seems new to me. The authors also tried hard to give readers enough intuition to understand what they are trying to achieve in this paper.

Weaknesses:

1. Although the authors tried to guide readers by intuitive explanation in the paper, they did this at the expense of sacrificing too much rigor and clarity in their statement of the theoretical results. For instance, Theorem 1 is too loose as a theorem. One might write a theorem like this in the introduction but not like this in the main text. Further, what does "using TNN constructed with $D \cup A$" mean in the statement of Proposition 2, given $D$ being the training and $A$ being a subset of the test data? As another example, the word "synapse" does not mean much to a machine learning person who knows nothing about neurology.

2. As new as TNN and SQANN look, I highly doubt if constructions alike are really helpful for interpretability issues. The type of interpretability that TNN and SQANN try to garner, at least to me, is the ability to memorize the training data and then tell if a new test data has a similar counterpart in the training data. I understand interpretability is not a well-defined mathematical concept so what it really means depends on the audience. But I do think the authors should write what "interpretable NN" means in this paper upfront as there are too many different types of interpretability people are talking about.

3. In the kind of interpretability that TNN and SQANN are pursuing, TNN only works for input in $\mathbb{R}$. But with input in $\mathbb{R}$, no matter how complicated the neural networks are, interpretability is not a big issue.

4. One has to constantly refer to the appendix even to understand how SQANN is constructed, the main algorithm of this paper.

5. No running-time analysis or empirical evidence is provided. The test examples are relatively simple (small data and low dimension). The main benefit of neural networks is the high prediction accuracy and scalability to high-dimension and big datasets. With this premise, interpretability then becomes important. Without showing that these new constructions can achieve state-of-the-art prediction accuracy in high-dimensional datasets, I do not see the importance of interpretability.

Further questions:

1. Is it necessary to write a whole paragraph on "Generalization to n-dimensions" for TNN? Maybe it is better to write one sentence and leave all the appendix as the authors did not think their solution is satisfactory in the TNN part.

2. If one uses SQANN in a one-dimensional setting, how does it compare to TNN? Does it also enjoy the properties that TNN has?

Some minor comments:

(1) It would be extremely helpful if the authors could create hyperlinks to all the sections/equations/figures that they refer to.

(2) In the statement of Lemma 1, "First layer of SQANN" should be "The first layer of SQANN".

(3) The appendix contains many typos and needs further proofreading.

**Summary Of The Paper:**

This paper proposes two new neural networks (NN) construction schemes that aim at better interpretability, in the sense that (1) the NN should always memorize the training data; (2) the NN can roughly tell if a new test data has any similarity to any data in the training sample. The authors also provide approximation error bounds on both training and test data under certain assumptions. Some numerical examples are shown to support their theory.

**Summary Of The Review:**

I do not doubt the theorems in this paper have any issues. But I do have trouble understanding the motivation of this work, as I have detailed in my weakness section. Without showing a learning algorithm achieves state-of-the-art prediction performance, I do not get why interpretability is of concern. As a result, I do not see how this work is a significant contribution to the ML community and would not recommend the paper to be published by this year's ICLR.

---

> ### Author Response · Authors · 2021-11-09
> **Author's Response**
>
> Thanks for taking the time to review the paper. We will revise our paper. The following is our attempts to address the feedback you have given.
>
> **1. Comment**: (1) “they did this at the expense of sacrificing too much rigor and clarity in their statement of the theoretical results”. (2) what does "using TNN constructed with $D\cup A$ " mean in the statement of Proposition 2,
>
> *Response*: (1) We don’t know which rigour is sacrificed: the theorem shows zero error clearly, and so do the examples we provide in the codes. (2) This means we reconstruct the TNN with not just D, but also with A: in another words, we have identified A as the data samples from test set that are out of distribution, and then include them into the model so they will be recognized in the future.
>
> **2. Comment**: I do think the authors should write what "interpretable NN" means in this paper upfront.
>
> *Response*: We do write it upfront. For TNN, it can be found in page 2, *ordered activation*, where we mention that one neuron is understood as the representative of one sample data point. For SQANN, the whole first chunk of paragraph in page 5, is where it can be found.
>
> **3. Comment**:  TNN only works for input in $\mathbb{R}$...
>
> *Response*: yes, we acknowledge that too, which is why SQANN is developed.
>
> **4. Comment**: One has to constantly refer to the appendix even to understand how SQANN is constructed
>
> *Response*: we are sorry about the page limit. We might try reordering things in the revised version.
>
> **5. Comment**: No running-time analysis or empirical evidence is provided. The test examples are relatively simple (small data and low dimension)
>
> *Response*: running time analysis for TNN is available, but for SQANN it will be highly dependent on dataset and the ordering of the data. The ordering of data for SQANN is for now out of the scope of this paper. Empirical evidence is available only as a small data for now, and, in further study, we expect to use improved version of SQANN for big data.
>
> **6. Comment**: The main benefit of neural networks is the high prediction accuracy and scalability to high-dimension and big datasets. With this premise, interpretability then becomes important.
>
> *Response*: very good point. In a sense, our construction tries to catch up with high-performing deep neural network from another direction. We believe post-hoc analysis and other methods to study the interpretability of high-performing neural networks have achieved a limited success, so we approach the research question from a slightly skewed perspective, if you like.
>
> **7. Comment**. “Is it necessary to write a whole paragraph on "Generalization to n-dimensions" for TNN?”
>
> *Response*: Reviewers are notoriously fickle. If we don’t show that we have considered the generalization, it might be cherry-picked as a lack of effort to pursue an obvious research direction.
>
> **8. Comment**. If one uses SQANN in a one-dimensional setting, how does it compare to TNN?
>
> *Response*: Unfortunately, we have not tried that explicitly. However, it should be clear that SQANN is more general. For example, after we order the dataset in 1 dimension, we can feed it into SQANN with different threshold parameters. Furthermore, with different way to perform interpolation in the case of weak activations, there may be a large room for exploration. That said, TNN itself can be modified, and many combinations will emerge. Things might then be considered case by case.

---

### Official Review · Reviewer_EKmc · 2021-11-04

**Correctness:** 2
**Technical Novelty And Significance:** 2
**Empirical Novelty And Significance:** 2
**Recommendation:** 3
**Confidence:** 4

**Main Review:**

The proposed methodology constructs the neural network according to the ordering of the input data and leaves a fingerprint for each data point in each neuron. No optimization is used in constructing the network. This is confusing and limits the applicability of the neural network.

The authors write that “a neuron in SQANN corresponds exactly to a data sample as SQANN stores its “fingerprints” as neurons’ nuclei”. In this case, how will the neural network be interpretable if it is trained on millions of data units? Will the proposed neural network still have millions of neurons and learned parameters? This questions interpretability. Also, how will the neural network conduct prediction for unseen test data?

The proposed methodology is very limited in dealing with high dimensional data. It is not clear how high dimensions can be incorporated in this modeling. This is an important issue that should not be overlooked.

Under the Subsection Generalizability to test dataset, are the authors pointing to using test data in training?

The experimentation is also limited and not convincing.

**Summary Of The Paper:**

This paper propose two new NN architectures, namely TNN, and SQANN. They claim that these networks are resistant to catastrophic forgetting, are interpretable, and are highly accurate.

**Summary Of The Review:**

There are several key issues about this paper that are not well-supported and are not clear.

---

> ### Author Response · Authors · 2021-11-09
> **Author's Response**
>
> Thanks for taking the time to review the paper. We will revise our paper. The following is our attempts to address the feedback you have given.
>
> **1. Comment**: The proposed methodology is very limited in dealing with high dimensional data
>
> *Response*: SQANN can be used for high dimensional data.
>
>
> **2. Comment**: Under the Subsection Generalizability to test dataset, are the authors pointing to using test data in training?
>
> *Response*: yes. See our response to reviewer KGST comment 5. In summary, we propose to include test datasets that are obviously out of distribution.
>
> **3. Comment**: In this case, how will the neural network be interpretable if it is trained on millions of data units? Will the proposed neural network still have millions of neurons and learned parameters?
>
> *Response*: ideally, if there 1% of the data unit are representative, we use only that 1% of data and do not need all 1 million data in our models. This is why we can train data on a fraction of the full dataset, and then use the property of our models to identify which other training data points are new (thus the data can be included into the updated model) or which data are already represented (thus the data can be regarded as validation data points). Otherwise, we can use other pre-processing methods to shave down the number of data samples. The interpretability remains the same: the data we input into SQANN can be identified with strongly activated neurons, or if it is out of distribution, we can find which neurons are activated most strongly i.e. which training samples are most closely related to this sample.

---

### Author Response · Authors · 2021-11-14
**Remarks, Clarifications, Our Revision**

## We understand that our review scores are very low, but here is the revision anyway.

Many thanks to the reviewers. We believe our revisions and clarifications can answer most of your doubts. Apart from this post, we respond to other comments in the respective author’s responses; do read them, since they are helpful too.

## Content Clarification
There are many possibly good points in the paper that seem to be missed out by reviewers. For example, (1) detection of ood samples is hardly mentioned throughout reviews (2) Reviewer v56a’s comments are almost exclusively based on TNN, though SQANN is a large part of our paper. We agree that some parts are not directly relevant, so we reorganize them and move some parts into appendix with additional explanations where necessary. We move things to appendix, for example based on Reviewer Xw97’s *further question 1* and reviewer v56a’s point 2.

**Concept disambiguation** section has been included in the introduction. We clarify the exact terms we use:
1. interpretability: *We consider only fine-grained interpretation, i.e. we look at the meaning of each single neuron or its activation in our models*. Thanks to Reviewer Xw97 comment “Weaknesses 2”.
2. catastrophic forgetting: *Readers might be familiar with universal approximation of functions on certain conditions…* We include reasons why ours will be more general. Thanks to Reviewer Reviewer v56a, “Weakness 7”.
3. universal approximation: *the tendency for knowledge of previously learned dataset to be abruptly lost as information relevant to a new dataset is incorporated*. In response to Reviewer v56a, we would like to again refute *The claimed “universal approximation” is not exactly the one machine learning cares*.

**Regarding Proposition 1**. Reviewer KGST seems to be somewhat bothered with the notion of universal approximation as a perfect fitting of training data. We assert that through perfect fitting and prediction mechanism e.g. SQANN interpolation, we are able to perform approximation. Reviewer probably wants to point out that standard ML training/validation/test practices should be followed strictly, which is understandable. However, this practice might have no resolution to out-of-distribution (ood) samples.

Therefore, we resolve this by adding a short paragraph before proposition 1 to explain how it works: some *subset of test datasets that are out-of-distribution* can be included into training dataset, so that the neural network learns previously unseen data. The paragraph relevant is our revised paper is ‘*With the following proposition, test dataset that resembles training dataset will yield small error. Otherwise, there are out-of-distribution (ood) samples…*’. What is *important to know here* is, for some existing ML methods, even when they are trained on test data to handle ood samples as we have done, they may STILL NOT PREDICT THEM CORRECTLY. Our method has clear advantage on this aspect, see appendix A2.2. *Advantage over existing methods* for more explanation.

**Using test dataset for training?** Yes, only AFTER we determine that they are out of distribution. This is to address most reviewers’ concern.

**Scalability** with good practice. We have included *Good practice for scalability* in appendix in response to comments on scalability, for example Reviewer EKmc’s comment ‘how will the neural network be interpretable if it is trained on millions of data units’. In essence, we choose a fraction of the data for construction, and then included more into the architecture when ood samples are found. We demonstrated this on the Boston and Diabetes Dataset.

**Reference**. We included [1] which visually demonstrated the intuition behind one way to construct universal approximation. Reviewer v56a says it *can be proven rigorously*; indeed, it can be, but the proof is not actually there.


[1] Nielsen, M. A. (2015). Neural networks and deep learning (Vol. 25). San Francisco, CA: Determination press.

## Minor Changes
Hyperlinks have been added as suggested by Reviewer Xw97.

---

### Decision · Program_Chairs · 2022-01-20

**Decision:**

Reject

**Comment:**

This paper propose two new neural network (NN) architectures, namely TNN, and SQANN. The paper claims that these networks are resistant to catastrophic forgetting, are interpretable, and are highly accurate. While the reviewers agree that the idea of making neurons reflect training data is novel, some concerns remain post rebuttal. Most of the reviewers opine that the statements of theorems are unclear, confusing, and hard to interpret (even after the rebuttal and update), thus making it hard to appreciate the contributions of this work. Given this, we are unable to recommend this paper for acceptance at this time. We hope the authors find reviewer feedback useful.